# Inertial Confinement Fusion Forecasting via Large Language Models

## Abstract

Controlled fusion energy is deemed pivotal for the advancement of human civilization. In this study, we introduce **LPI-LLM**, a novel integration of Large Language Models (LLMs) with classical reservoir computing paradigms tailored to address a critical challenge, Laser-Plasma Instabilities (LPI), in Inertial Confinement Fusion (ICF). Our approach offers several key contributions: Firstly, we propose the *LLM-anchored Reservoir*, augmented with a *Fusion-specific Prompt*, enabling accurate forecasting of LPI-generated-hot electron dynamics during implosion. Secondly, we develop *Signal-Digesting Channels* to temporally and spatially describe the driver laser intensity across time, capturing the unique characteristics of ICF inputs. Lastly, we design the *Confidence Scanner* to quantify the confidence level in forecasting, providing valuable insights for domain experts to design the ICF process. Extensive experiments demonstrate the superior performance of our method, achieving 1.90 CAE, 0.14 `top-1` MAE, and 0.11 `top-5` MAE in predicting Hard X-ray (HXR) energies emitted by the hot electrons in ICF implosions, which presents state-of-the-art comparisons against concurrent best systems. Additionally, we present LPI4AI, the first LPI benchmark based on physical experiments, aimed at fostering novel ideas in LPI research and enhancing the utility of LLMs in scientific exploration. Overall, our work strives to forge an innovative synergy between AI and ICF for advancing fusion energy.

## 1 Introduction

> *"...human society remains at a Type 0, a primitive form of civilization..."*
>
> − The Kardashev scale (Kardashev, 1964)

After National Ignition Facility (NIF) achieving ignition in December 2022 (Abu-Shawareb et al., 2024), the focus of current inertial confinement fusion (ICF) research shifts to exploring high gain schemes required to make fusion a practical and sustainable energy source for humankind. Fusion represents a potential key enabler for advancing humanity towards a Type I civilization on the Kardashev scale (Zhang et al., 2023), offering a virtually limitless and clean energy source that could power our civilization globally. This advancement could potentially resolve numerous crises we currently face — *e.g.*, economic recessions and climate change — by eliminating the need for finite resources like fossil fuels.

Direct-drive ICF has potentially higher gains due to more efficient driver-target coupling but faces many challenges (Betti & Hurricane, 2016). The optimization of ICF designs to achieve reliable high-gain ignition faces formidable constraints (Betti & Hurricane, 2016; Craxton et al., 2015) due to laser-plasma instabilities (LPI) (Gopalaswamy et al., 2024; Radha et al., 2016). Efficient and symmetrical driving of the target, vital for ICF, is impeded by LPI phenomena such as stimulated Raman and Brillouin backscatterings (SRS and SBS), which can disrupt implosion symmetry and reduce efficiency through cross-beam energy transfer (CBET) (Smalyuk et al., 2008a; Goncharov et al., 2008). Hot electrons, a byproduct of LPI processes like SRS and Two-Plasmon-Decay (TPD), can both hinder (Smalyuk et al., 2008a; Goncharov et al., 2008; Craxton et al., 2015; Radha et al., 2016) and assist (Betti et al., 2007; Perkins et al., 2009; Shang et al., 2017) ignition, showcasing the importance of LPI. Despite efforts to measure and simulate hot electron generation, obtaining predictive scaling laws remains challenging due to the dynamic nature of laser/plasma conditions

and computational limitations (Klimo et al., 2010; Riconda et al., 2011; Yan et al., 2014; Shang et al., 2017; Li et al., 2020), highlighting the current gap in establishing a predictive capability based on first principles that aligns with experimental data. These constraints highlight the need for an innovative approach to overcome these obstacles.

Currently, Large Language Models (LLMs) exhibit versatile capabilities across diverse disciplines (*e.g.*, robotics (Lin et al., 2022; Szot et al., 2021; Yao et al., 2023; Huang et al., 2022), medical health (Li et al., 2023; Singhal et al., 2023; Thirunavukarasu et al., 2023; Moor et al., 2023), agriculture (Rezayi et al., 2022; Tzachor et al., 2023)), adeptly capturing intricate patterns in multimodal data. Due to their success in other domains (Kojima et al., 2022; Zheng et al., 2023; Liu et al., 2023), we are convinced that LLMs may also potentially excel in generalizing to plasma physics, particularly in forecasting the behavior of

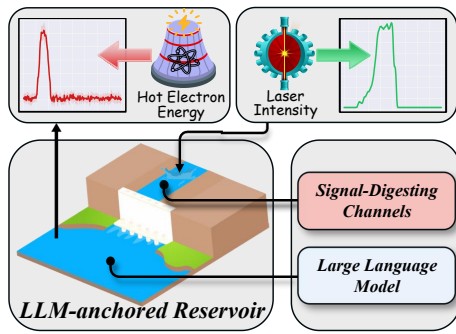

Figure 1: The overview of **LPI-LLM**.

hot electrons generated during implosion in the real-world, physics experiments. Leveraging their vast pre-trained knowledge base, LLMs could optimize `ICF` designs by efficiently evaluating numerous scenarios, aiding researchers in identifying promising approaches expediently, and generating insights to enhance our understanding and control of the fusion process.

In light of this perspective, we intend to harness the power of LLMs to overcome the barrier to the vertical advancement of the next stage of human civilization: fusion energy. This science problem manifests as two sub-problems in a unique and challenging setup: ❶ how to *tailor pre-trained LLMs* in order to accurately predict the behavior of hot electrons based on laser intensity inputs? and ❷ how to evaluate the *trustworthiness* of the LLM's predictions in order to guide the `ICF` design?

We conceptualize LLMs as a computational reservoir to unlock their potential for robust domain adaptation and generalization for `ICF` task, titled **LPI-LLM** (see Fig. 1). In order to address question ❶, we propose the *LLM-anchored Reservoir*, augmented with a fusion-specific prompt, to facilitate the interpretation of plasma physics. The incorporated prompts encompass domain-specific knowledge, instructional cues, and statistical information, thereby enabling the LLMs to accommodate the specific demands of the given task. Additionally, we develop the *Signal-Digesting Channels* to capture the distinctive characteristics of `ICF` inputs. It features a temporal encoder to better align the laser signals with the pre-trained, time-series space, and a spatial encoder to provide a global description of the input landscape. To tackle question ❷, we introduce the *Confidence Scanner* to assess the trustworthiness in predictions. Specifically, we couple the gradient saliency in prediction head and token entropy in LLM outputs to obtain the model's confidence scores.

Overall, this study aims to provide an exciting synergy between the domain of AI and plasma science for fusion energy development. The core contributions of this paper can be summarized as follows:

- *We represent a **pioneer investigation** into the use of LLMs for analyzing hot electron dynamics in `ICF`.* LLMs offer a cost-effective alternative compared to both `ICF` experiments and plasma-physics simulations. Empirical evidence (see §3/S3) showcases the efficacy of the application of LLM for `LPI` study, which directly benefits the practical design of `ICF`.
- *We construct an **LLM-anchored reservoir computing** framework to predict the hot electron dynamics in `ICF` implosions.* Compared to prior arts in reservoir computing (Li et al., 2024; Gauthier et al., 2021) and time-series-based LLM (Jin et al., 2023), our approach requires less data (see Table 2d) and shorter training schedule (see Table 2e), while achieving superior performance.
- *We develop the **first `LPI` benchmark** — LPI4AI (see §3.1) based on physical experiments*, to facilitate new ideas in `LPI` research and the use of LLMs in scientific exploration.

## 2 METHODOLOGY

### 2.1 PRELIMINARY

**The ICF Overview.** `ICF` is one of the two major branches of fusion energy research. The main idea of `ICF` is using intense

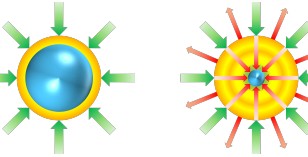

Figure 2: The **ICF** pipeline.

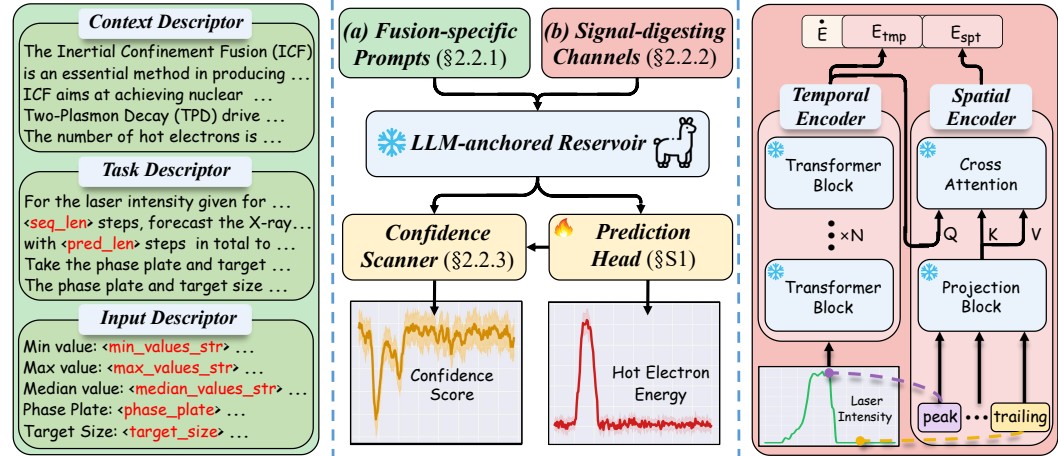

Figure 3: **LPI-LLM framework**. (a) *Fusion-specific prompts* structure domain textual prompts with context, task and input descriptions (see §2.2.1 for details). (b) *Signal-digesting channels* comprise a pre-trained temporal encoder to extract time-series features of laser signals, and a spatial encoder to encode critical landscapes of the inputs (see §2.2.2 for details). For simplicity, we skip some architectural modules. We provide implementation details in appendix (see §S2).

drivers to compress the target and maintain the fusion fuel at fusion densities and temperatures with its own inertia, which facilitates the occurrence of fusion reactions. The process of a conventional direct drive `ICF` is depicted in Fig. 2. Laser beams ➡ are directed towards a target filled with fuel 🔵, rapidly heating its surface to create a plasma envelope 🟡. The target surface will eventually blow off, causing the target shell to implode toward the center via the rocket effect. The kinetic energy of the shell will be converted into thermal energy of the fuel, which sustains the fuel at high densities (more than twenty times that of lead) and temperatures (approximately 100,000,000°C), thus initiating fusion reactions. Throughout this process, laser plasma instabilities (`LPIs`) could take energy from the laser and generate hot electrons, which may prematurely heat the target and emit Hard X-Rays (`HXR`) via bremsstrahlung radiation ➡ as they interact with the surrounding plasma. `HXR` diagnostics have been extensively applied in experiments at OMEGA (Smalyuk et al., 2008b; Yaakobi et al., 2000; Stoeckl et al., 2003; Yaakobi et al., 2005) and NIF (Rosenberg et al., 2018; 2020; Solodov et al., 2020) to determine the hot electron energy and temperature. It is worth noting that conducting these experiments is expensive (around **$1m** for one successful NIF shot). Having a predictive tool will not only improve our understanding of hot electron physics, but also benefit the design of `ICF`.

**Experiment and Data Collection.** The HXR data were collected from 100 shots conducted on the OMEGA platform. Detailed configurations, including the target size, the phase plate shape for laser smoothing, and the input laser profile, were available. `HXR` signals were recorded from four diagnostic channels, enabling the calculation of the total energy and temperature of the electrons. In this study, we focus on forecasting `HXR` signals with given laser intensity profiles throughout the shot. This data is indispensable for physicists to design new `ICF` experiments.

## 2.2 LPI-LLM

In this section, we introduce LPI-LLM, a novel approach for predicting `LPI` and hot-electron generation in `ICF`. Illustrated in Fig. 3, the inputs comprise fusion-specific prompts and the time series of input laser intensity, which are processed through LLM for feature extraction. Subsequently, the output module makes predictions of hot-electron energy along with confidence scores. Our approach is characterized by three core modules: *LLM-anchored Reservoir*, which establishes a reservoir foundation using LLMs to comprehend the dynamic impact of laser intensity on hot-electron energy emission; *Signal-Digesting Channels* is responsible for encoding the time series of input laser intensity, capturing the temporal characteristics of sequential details, and spatial distinctiveness of data landscapes; and *Confidence Scanner*, tasked with estimating prediction confidence for each shot, thereby providing trustworthy guidance for practical experimental design of the `ICF`.

### 2.2.1 LLM-ANCHORED RESERVOIR

In classic reservoir computing (RC), a fixed, randomly generated reservoir (*e.g.*, RNN) transforms input data into a high-dimensional representation. A trainable output layer then maps this representation to the desired output. RC has gained popularity in the scientific community due to its ability to enable efficient processing of sequential data with simple training methods. Following the RC convention, we construct a reservoir using LLM with a shallow prediction head (see §S2). Due to the extensive pre-training, LLMs are equipped with robust generalization capabilities for in-context reasoning and time-series forecasting. To leverage the physics knowledge embedded in LLMs, we design *fusion-specific prompts* (FSP) that strategically connect the LLM's vast knowledge base to the specific nuances of the ICF domain. Our LLM-anchored reservoir can be defined by:

$$s[t + 1] = f(s[:t], E_{las}, {}^{**}\texttt{Prompt}), \qquad (1)$$

where $s[t]$ denotes the state of the reservoir at time step $t$ and $E_{las}$ represents input laser intensity.

The $^{**}$`Prompt` specifically denotes the structured FSP, comprising three descriptors (see Fig. 3a) designed with particular focus: ➤ *Context descriptor* provides a detailed overview of the ICF process, highlighting the nature, sources, and characteristics of the data. It elaborates on the experimental procedures (see §2.1) used in data collection, emphasizing principles and methodologies specific to ICF. This descriptor enhances the LLMs' comprehension of the experimental context. ➤ *Task descriptor* outlines explicit instructions for the prediction tasks, including the format and expected length of the output. It guides the inference and forecasting process by specifying imperative considerations, ensuring that the forecasting aligns with domain-specific insights. ➤ *Input descriptor* presents a concise statistical summary of the input data, offering insights into its distribution and key characteristics such as minimum, maximum, and median values. This descriptor is vital for informing the LLMs about the underlying statistical properties of the input signals. Collectively, this prompting strategy facilitates the LLMs' ability to mine intricate input signals and produce predictions that are scientifically robust and contextually coherent within the phase space of the reservoir.

In general, our LLM-based design expands the applications and capabilities of reservoir computing, offering two significant advantages in leveraging artificial intelligence for scientific exploration:

- *Adaptability for fusion.* The utilization of LLMs in ICF forcasting exhibits notable adaptability in confronting the pivotal scientific challenge – modeling LPI and hot electron generation. Later, we will present empirical evidence to substantiate the systemic efficacy (see §3.1). The success of this methodology is poised to provide a generic alternative for other adjacent scientific domains in pursuit of LLM-based solutions.

- *Efficiency for data scarcity.* Gratitude is extended for the extensive pre-trained knowledge base and the precise delineation of prompt descriptors. Leveraging these assets, the LLM-based system mitigates data dependency to the $K$-shot level — 80 shots in our experiments — exemplifying the advantageous efficiency in addressing the persistent challenge of data scarcity in scientific inquiry.

### 2.2.2 SIGNAL-DIGESTING CHANNELS

With the establishment of the LLM-based approach described in §2.2.1, robust predictions can be immediately attained (see Table 2c). Due to the uniqueness of the ICF data, the input HXR signal exhibits its time-series, temporal landscape. Recognizing the sensitivity of LLMs to input data (Sclar et al., 2023; 2024), we introduce *Signal-Digesting Channels* (SDC), conceptually aligned with our design, to capture crucial input characteristics and further augment the performance of LLMs.

For the ICF process, the hot electron energy emitted during the initial and terminal phases is characterized by relatively uniform values, in contrast to the target impact phase, where peak values are observed. This laser intensity signal landscape exhibits significant discrepancies between the uniformity of the plain phase and the peaks of the impact phase. This physics insight guides our design (see Fig. 3b), which comprises two components: a *temporal encoder* to align the laser intensities with the pre-trained time-series space, and a *spatial encoder* to delineate the landscape of input data.

➤ *Temporal encoder* is designed to extract time-series features using a windowing mechanism that constructs consistent signal patches across sequential time steps. It employs a set of Transformer layers (Woo et al., 2024) to capture time-series distributions over the forecast horizon. This process is formulated as $f_{tmp} : (X_{t-l:t+h}, Y_{t-l:t}) \mapsto \hat{\psi}$, where $X$ and $Y$ represent the input data and target

data spanning time windows of length $l$ and $h$ at time $t$. The encoder is pre-trained on a large-scale time-series dataset (see §S2 for details) and is optimized using the log-likelihood of the forecast:

$$\arg\max_{\theta} \quad \mathbb{E}_{(X,Y)\sim p(D)} \log p(Y_{t:t+h}|f_{tmp}(X_{t-l:t+h}, Y_{t-l:t})),$$

where $p(D)$ represents the data distribution from which the time-series samples are drawn. During fine-tuning on the target-domain `ICF` data, all pre-trained parameters are frozen, except for the last linear layer. The temporal encoder is utilized for feature extraction over the input laser signals $I$, producing temporal tokens denoted as $E_{tmp} = f_{tmp}(I)$ for subsequent processing.

➤ *Spatial encoder* is designed to analyze the input signals by providing a qualitative overview of the input landscape. Specifically, it is structured to characterize the spatial patterns of laser intensity signals throughout the `ICF` process. We utilize the projection block from the LLM, $f_{LLM}$, to project sets of critical contexts into spatial features, denoted as $E_{spt} = \{f_{LLM}(\text{"pulse"}), f_{LLM}(\text{"peak"}), ..., f_{LLM}(\text{"trailing"})\}$. In practice, $E_{spt}$ is further processed by a cross-attention layer with temporal features $E_{tmp}$. This step couples the contextual description to the actual signal distribution within the `ICF`, enabling the LLM to better capture the observed physical phenomena for predictions.

After acquiring the features from both the temporal and spatial encoders, we concatenate them $\dot{E} = [E_{tmp}; E_{spt}]$ to form the output of DSC. Here, we use $\dot{E}$ as augmented inputs to replace the vanilla $E_{las}$ in Eq.1. Fundamentally, DSC introduces a novel method for input encoding in reservoir computing, which enhances overall system performance. This design offers the following advantages:

- *Discernment for temporal pattern.* With the pre-trained temporal encoder, SDC adeptly captures crucial time-series features of laser signals that correlate with `HXR` outputs. This merit enables the LLM to recognize distinct patterns across various time steps, representing an improvement over strong reservoir models (see Table 2c), which struggle to manage `ICF`'s temporal patterns.

- *In-context reasoning in signal processing.* SDC tackles the complexity of processing signals with diverse attributes, such as those found in the uniform and peak phases within `ICF`. Through the integration of contextual disciplinary knowledge, SDC, equipped with in-context reasoning, significantly boosts the effectiveness of LLM backbone (see Table 2c). This enhancement enables robust performance even in the face of inherent fluctuations (*i.e.*, uniform *vs.* peak in `ICF`).

### 2.2.3 CONFIDENCE SCANNER

Trustworthiness is pivotal for AI in science. Under-confident predictions may lead to misguided conclusions or improper actions. Some approaches (Lin et al., 2023; Huang et al., 2023) directly utilize the entropy observed in the output tokens of the LLMs to gauge the confidence. However, these methods encounter a challenge in our study of `ICF` whereby the entropy of LLM's

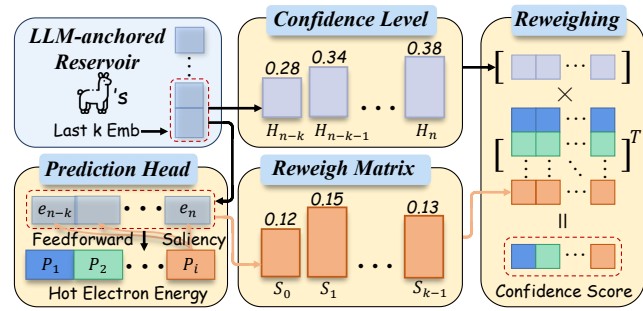

Figure 4: **Pipeline of Confidence Scanner**. We re-calibrate token confidence to align with the energy-prediction head.

output tokens does not consistently reflect confidence of prediction at each time step. This discrepancy arises due to the non-linear transformation undertaken by the multi-layer perception within the prediction head, which alters the embedding of token counterparts, thereby distorting the relation between tokens and their corresponding confidence estimations.

To this end, we propose *Confidence Scanner* that incorporates a confidence reweighing mechanism to assess the confidence level of each prediction systematically. Concretely, our approach redistributes confidence across tokens to implicitly reflect their actual influence on predictions.

As shown in Fig. 4, we extract the embedding $E_k = \{e_{n-k}, \ldots, e_n\}$ for the last layer of LLM, which specifically analyze the embedding of the last $k$ tokens. The confidence level $H$ is then formulated as:

$$H = [h(e_{n-k}), \ldots, h(e_n)], \tag{2}$$

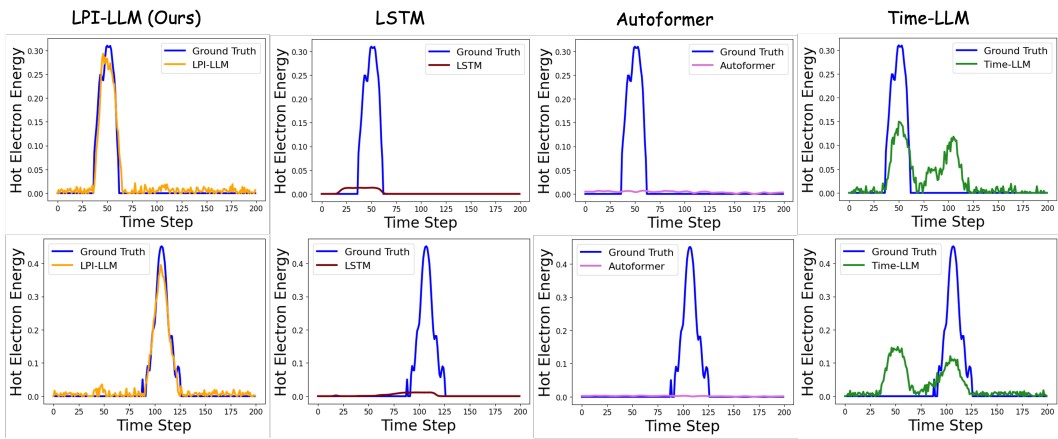

Figure 5: Qualitative results of 2 **hot electron prediction** cases. We plot Ground Truth and the predictions of Ours, Autoformer, Time-LLM and LSTM. Y and X axes denote hot electron energy in voltage, and time steps with step length of 0.025 nanosecond respectively.

where $h(\cdot)$ map the embedding to word probabilities, which are subsequently used to calculate the entropy. Furthermore, we derive a reweigh matrix $S$ from the task prediction head $H_{task}(\cdot)$ by performing a forward process to predict hot-electron energy by $P = H_{task}(E_k)$. The matrix $S$ is then obtained through saliency, reflecting the contribution to the $i$-length of predictions:

$$S = \left[ \sigma \left( \frac{\partial P_0}{\partial E_k} \right), \ldots, \sigma \left( \frac{\partial P_i}{\partial E_k} \right) \right], \tag{3}$$

where the $\sigma$ is the softmax function to normalize the saliency scale. Finally, entropy $H$ is combined with reweigh matrix $S$ to produce the confidence score $C = -H \times S$ for the hot electron energy predictions. Through our design, we align the entropy derived from LLM embeddings with the hot electron energy forecasting. This alignment allows us to directly obtain a confidence level that can serve as a trustworthiness indicator for our system. We provide empirical evidence in Fig. 6.

## 3 EMPIRICAL FINDINGS

### 3.1 MAIN RESULTS

**Dataset.** We develop a new benchmark, LPI4AI, to support AI research in `ICF`. This benchmark consists of 40,000 `LPI` samples containing 100 shot sequences with 400 time steps/shot. Each shot is documented with key parameters such as target size, laser intensity, and energy of hot electrons (see §2.1). The dataset has been systematically divided into 80/10/10 for `train`/`val`/`test` splits respectively. We will release this dataset upon acceptance to advance research for the fusion reaction.

**Baselines.** We choose a classic physics-based Particle-In-Cell (PIC) simulation method (Cao et al., 2022), a number of classic AI models (*i.e.*, LSTM (Hochreiter & Schmidhuber, 1997), Autoformer (Wu et al., 2021)), reservoir computing models (*i.e.*, HoGRC (Li et al., 2024), RCRK (Dong et al., 2020), NGRC (Gauthier et al., 2021)), and concurrent time-series LLM-based models (*i.e.*, GPT4TS (Zhou et al., 2023) and Time-LLM (Jin et al., 2023)) as baseline models for performance comparison on proposed LPI4AI dataset.

**Experimental Setup.** Our experiments are trained with 100 epochs and a batch size of 5, which is adequate to achieve convergence based on our empirical findings. In addition, we utilize a fixed learning rate of $0.0004$ and the Adam optimizer (Kingma & Ba, 2015). A loss function defined by the cumulative absolute error across each time-steps $f_{loss}(Y, G) = \sum_{n=1}^{pred\_len} |y_n - g_n|$ is used, where $Y$ and $G$ represent the sequences of predictions and ground truths, respectively, $y_n$ and $g_n$ denote the values at the $n$-th time-step, and pred_len is the length of prediction. For all other counterpart methods, we follow their original settings and training configurations to reproduce the results.

**Metric.** We employ cumulative absolute error (CAE) as our primary metric. The sole distinction in its implementation involves nullifying values that are less than 0.03 of the predicted value. In

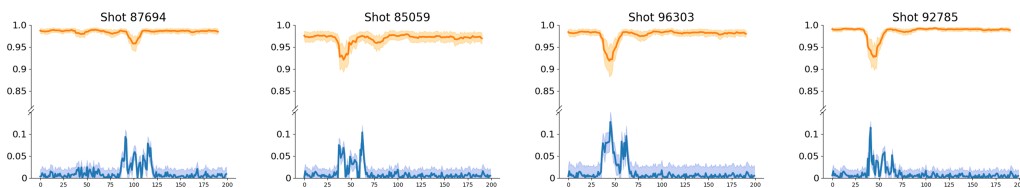

Figure 6: Visualization of **confidence score** and **prediction error**.

addition, we incorporate two supplementary metrics: `top-1` and `top-5` MAE, which represent mean absolute error focusing exclusively on the top one percent or five percent errors, respectively, thereby highlighting the performance where the highest errors are observed.

**Performance Comparison.** The primary challenge in forecasting for `LPI` lies in developing a robust yet flexible predictive model. We integrated four wellk-known open-sourced LLMs with our apporach, including Gemma-2 (Team, 2024), OLMo (Groeneveld et al., 2024), Llama-2 (Touvron et al., 2023b), and Llama-3 (AI, 2024). As shown in Table 1, empirical results demonstrate the effectiveness

Table 1: **Quantitative results** on LPI4AI `test` split for hot electron energy predictions (see §3.1 for details). Refer to the metrics section for details of CAE, `top-1` and `top-5` MAE.

| Method | CAE↓ | top-1 MAE↓ | top-5 MAE↓ |
|---|---|---|---|
| PIC Simulation | 2.88 | 0.20 | 0.13 |
| LSTM | $5.82_{\pm0.06}$ | $0.35_{\pm0.01}$ | $0.35_{\pm0.01}$ |
| Autoformer | $5.79_{\pm0.04}$ | $0.35_{\pm0.01}$ | $0.34_{\pm0.01}$ |
| HoGRC | $4.20_{\pm0.79}$ | $0.25_{\pm0.05}$ | $0.22_{\pm0.02}$ |
| RCRK | $4.31_{\pm0.46}$ | $0.28_{\pm0.04}$ | $0.22_{\pm0.01}$ |
| NGRC | $4.28_{\pm0.68}$ | $0.27_{\pm0.04}$ | $0.23_{\pm0.02}$ |
| GPT4TS | $3.34_{\pm0.58}$ | $0.18_{\pm0.05}$ | $0.14_{\pm0.04}$ |
| Time-LLM | $3.48_{\pm0.72}$ | $0.18_{\pm0.05}$ | $0.15_{\pm0.05}$ |
| **LPI-LLM (Gemma-2-9B)** | $2.04_{\pm0.21}$ | $\mathbf{0.14}_{\pm0.01}$ | $0.12_{\pm0.01}$ |
| **LPI-LLM (OLMo-7B)** | $1.97_{\pm0.28}$ | $\mathbf{0.14}_{\pm0.01}$ | $0.12_{\pm0.01}$ |
| **LPI-LLM (Llama-2-7B)** | $2.15_{\pm0.26}$ | $\mathbf{0.14}_{\pm0.01}$ | $0.12_{\pm0.01}$ |
| **LPI-LLM (Llama-3-8B)** | $\mathbf{1.90}_{\pm0.33}$ | $\mathbf{0.14}_{\pm0.01}$ | $\mathbf{0.11}_{\pm0.01}$ |

of our method in predicting hot electron energy output in `ICF`, consistently outperforming all baseline models across all evaluation metrics. Notably, our approach over Llama-3 achieves the best performance and surpasses PIC simulation model (Cao et al., 2022) by **0.98** in terms of CAE. Additionally, compared to classic AI methods, our approach outperforms LSTM (Hochreiter & Schmidhuber, 1997) and Autoformer (Wu et al., 2021) by **3.92** and **3.89**, respectively. Our margins in CAE over three other reservoir computing methods, HoGRC (Li et al., 2024), RCRK (Dong et al., 2020), NGRC (Gauthier et al., 2021), are **2.30**, **2.41**, and **2.38**, respectively. Moreover, our method outperforms concurrent LLM-based approaches GPT4TS (Zhou et al., 2023) and Time-LLM (Jin et al., 2023) by **1.44** and **1.58** on CAE, respectively, while maintaining comparable speed. Detailed analysis is supplemented in §S3.

These results underscore the efficacy and efficiency of our LLM-based solution in predicting hot electron dynamics in `ICF`. By extending LLM's successful adaptability to the new and exciting domain of fusion energy, our empirical findings represent just the beginning of the innovative opportunities presented by applying LLM algorithms to challenging subjects in scientific exploration.

We further present qualitative results in Fig. 5, aligning with our quantitative findings that our method surpasses all comparative baselines in predictive accuracy. Additionally, Fig. 6 illustrates the confidence scores associated with our predictions. The visualization elucidates a clear correlation between predictive error and confidence scores, indicating high confidence corresponding to low errors and conversely. Notably, our approach consistently demonstrates a heightened level of confidence, particularly in forecasting peak values across the sequence, a critical phase in `ICF`.

## 3.2 DIAGNOSTIC EXPERIMENT

This section ablates LPI-LLM's systemic design on `val` split of LPI4AI. All experiments use the Llama 3 8B variant. **Appendix §S3 has more experimental results.**

**Key Component Analysis.** We first investigate the two principal modules of LPI-LLM, specifically, Fusion-Specific Prompt and Signal-Digesting Channels. We construct a baseline model with generic dummy prompts which only provide broad, non-specific instructions regarding fusion, and a rudimentary encoder composed of a single linear layer. As shown in Table 2a, the baseline model achieves 3.57 CAE. Upon applying Fusion-Specific Prompt to the baseline, we observe significant

improvements for CAE from 3.57 to **2.59**. Furthermore, after incorporating Signal-Digesting Channels into the baseline model, we achieve significant gains of **1.56** CAE. Finally, by integrating both core techniques, our LPI-LLM delivers the best performance of **1.19** CAE. These findings affirm that the proposed Fusion-Specific Prompt and Signal-Digesting Channels operate synergistically, and validate the effectiveness of our comprehensive model design.

**Fusion-Specific Prompt.** We next study the impact of our Fusion-Specific Prompt by contrasting it with a constructed baseline. This baseline incorporates Signal-Digesting Channels and employs generic prompts that provide broad, nonspecific instructions unrelated to the process of fusion. As shown in Table 2b, the baseline yields a performance measure of 2.01 in terms of CAE. Upon substituting the generic prompt with one that integrates discipline-specific information, including background knowledge, task

Table 2: A set of **ablative studies** evaluated on `val` split.

(a) Key Component Analysis

| Algorithm Component | CAE↓ |
|---|---|
| BASELINE | 3.57 |
| + Fusion-Specific Prompt | 2.59 |
| + Signal-Digesting Channels | 2.01 |
| **OURS** (**both**) | 1.19 |

(d) Different # of Samples

| # of samples | **Ours** | HoGRC | NGRC | Time-LLM |
|---|---|---|---|---|
| 80 | 1.19 | 3.80 | 3.73 | 3.60 |
| 60 | 1.73 | 3.87 | 3.84 | 3.69 |
| 40 | 2.56 | 4.01 | 4.08 | 3.94 |
| 20 | 3.47 | 4.35 | 4.19 | 4.10 |

(b) Fusion-specific Prompt

| Prompt Type | CAE↓ |
|---|---|
| BASELINE | 2.01 |
| + Discipline-related Prompt | 1.46 |
| + Input Statistics | 1.58 |
| **OURS** (**both**) | 1.19 |

(e) Different # of Epochs

| # of epochs | **Ours** | HoGRC | NGRC | Time-LLM |
|---|---|---|---|---|
| 100 | 1.19 | 3.80 | 3.73 | 3.60 |
| 50 | 1.19 | 3.99 | 3.83 | 3.60 |
| 20 | 1.56 | 4.12 | 4.35 | 4.01 |
| 10 | 2.79 | 5.23 | 5.50 | 4.52 |

(c) Signal-digesting Channels

| Algorithm Component | CAE↓ |
|---|---|
| BASELINE | 2.59 |
| + Temporal Encoder | 2.47 |
| + Spatial Encoder | 1.41 |
| **OURS** (**both**) | 1.19 |

(f) Different Head Dimension

| Head Dim. | # Params | CAE↓ | top-5 MAE↓ |
|---|---|---|---|
| 256 | 102.4 K | 1.23 | 0.12 |
| 128 | 51.2 K | 1.19 | 0.11 |
| 64 | 25.6 K | 1.34 | 0.11 |
| 32 | 12.8 K | 1.57 | 0.13 |

instructions, *etc*, there is an observable enhancement in performance, achieving an improvement of **0.55** in CAE over the baseline. Additionally, a further analysis involving the integration of input statistics, containing the maximum and minimum values, *etc*, of the input time series, demonstrates superior performance, outperforming the baseline by **0.43** in CAE. The most notable enhancement is recorded when employing our Fusion-Specific Prompt, which amalgamates both the discipline-related information and input statistics, culminating in a peak performance of **1.19** CAE. This outcome highlights the essential function of the Fusion-Specific Prompt within our approach, significantly impacting the performance of the overall model.

**Signal-Digesting Channels.** We then examine the influence of Signal-Digesting Channels in Table 2c. For the baseline, we use a basic approach comprising solely a single linear layer. Under this setting, the baseline model achieves 2.59 in terms of CAE. Integration of either the Temporal Encoder or the Spatial Encoder independently results in performance improvements of **0.12** and **1.18** above the baseline respectively. Conversely, the integration of both Encoders in SDC substantially surpasses all alternative counterparts, achieving a CAE of **1.19**. These results substantiate the hypothesis that the proposed Signal-Digesting Channels augment the capability of our approach to more accurately interpret input time series data.

**Reservoir and LLM Comparison.** To thoroughly explore the training effectiveness of our methodology under conditions of limited sample availability, we perform a comparative analysis using a variable number of training samples with two concurrent reservoir methods (Li et al., 2024; Gauthier et al., 2021) and one LLM-based method (Jin et al., 2023) in Table 2d using CAE. The empirical findings demonstrate that our approach consistently outperforms all competing strategies across various sample configurations. Notably, this superior performance and training effectiveness are evident even with as few as 20 samples. Such robust efficacy is critical in scientific AI applications, where datasets are often constrained in size.

In addition, we study the training efficiency of our approach in contrast to the above strong baselines (Li et al., 2024; Gauthier et al., 2021; Jin et al., 2023) in Table 2e, across various training epochs. The experimental outcomes illustrate that our approach not only outperforms its counterparts but also demonstrates superior efficiency. Specifically, our method is capable of achieving comparable or superior performance in a significantly reduced training duration. For instance, our model requires only 10 epochs to achieve better performance than other methods that require 100 epochs. This enhanced efficiency in training is particularly significant, as it demonstrates the potential of our approach to deliver robust performance swiftly, thereby facilitating more expedient research in practical scenarios.

**Prediction Head Dimension.** Lastly, we conduct additional experiments to evaluate the impact of varying dimensions in the prediction head. As shown in Table 2f, our approach demonstrates an enhancement in CAE, reducing from 1.57 to **1.34**, concomitant upon augmenting the head dimension from 32 to 64. This improvement continues, culminating in a CAE of **1.19** at a head dimension of 128, where it stabilizes, indicating this as the optimal head dimension for balancing effectiveness and parameter-efficiency. We therefore select the dimension of 128 as the default setting.

## 4 RELATED WORK AND DISCUSSION

**AI for Science.** AI has increasingly become a vital tool in advancing scientific discovery, playing a central role in recent breakthroughs across various fields (Jumper et al., 2021; LeCun et al., 2015; Reichstein et al., 2019; Ali et al., 2024). The trajectory of AI's involvement in scientific research began with elementary data analysis techniques, such as rule-based systems (Breiman, 2001; Safavian & Landgrebe, 1991), Bayesian methods (Frank et al., 2000), analogy-based approaches (Hearst et al., 1998; Jain et al., 1999; Tenenbaum et al., 2000), evolutionary algorithms (Kennedy & Eberhart, 1995; Dietterich, 2000), and connectionist models (Weisberg, 2005; LeCun et al., 2015). These methods laid the foundation for AI's contributions to scientific exploration, and have since evolved into sophisticated models, including deep learning (He et al., 2016), transformers (Vaswani et al., 2017), and foundation models (Dosovitskiy et al., 2021; Bommasani et al., 2021). These advancements have empowered scientists to tackle complex problems with greater accuracy and efficiency.

Distinct from traditional AI approaches in scientific exploration, our work represents a one of the first efforts to empower a crucial scientific domain in ICF with LLMs, which has previously relied on traditional computational techniques. Despite lacking explicit first-principle rules, our LLM-based models exhibit remarkable predictive accuracy on real-world data, offering a promising alternative to traditional simulations and costly experimental data collection. Specifically, our method demonstrates the potential to revolutionize processes such as ICF, by providing reliable, data-driven insights that guide experimental setups with reduced reliance on conventional computation methods.

**Plasma Physics for Fusion.** LPI is an important area of study within plasma physics for fusion due to its potential to decrease the efficiency of the implosion process. Stimulated Raman scattering (SRS) and stimulated Brillouin scattering (SBS) can cause the reflection of laser beams (Kirkwood et al., 1999). Cross-beam energy transfer (CBET) can adversely influence the symmetry of implosion (Igumenshchev et al., 2010). Two-plasmon decay (TPD) can effectively generate hot electrons, which can preheat the target, increase the shell entropy, and diminish the implosion efficiency. (Smalyuk et al., 2008a; Goncharov et al., 2008; Craxton et al., 2015; Radha et al., 2016). In shock ignitions (Betti et al., 2007; Perkins et al., 2009), hot electrons can also deposit their energy in the compressed shell, thereby enhancing the ignition shock and aiding ignition processes. Understanding LPI physics and establishing predictive models for hot electron generation in direct drive ICF are crucial. In response to these issues and challenges, extensive experiments and simulations (Smalyuk et al., 2008a; Goncharov et al., 2008; Craxton et al., 2015; Radha et al., 2016; Betti et al., 2007; Perkins et al., 2009) are necessary, which are both costly and time-consuming.

To address these challenges, our AI-based approach offers a promising alternative. By implementing a streamlined LLM pipeline, our model acts as a Computational Reservoir for time-series forecasting, capturing domain-specific knowledge and generalizing within the ICF task. This enables LLMs to assist in ICF design, reducing reliance on costly experiments and simulations. Empirical results (see §3) suggest that LLMs could revolutionize predictive modeling in plasma physics, providing rapid, cost-effective insights based on generalizable knowledge.

Notably, our LPI-LLM for predicting hot electron-induced hard X-rays would provide a useful framework for predicting other experimental data, such as neutron yields, with different but equally intricate underlying physics. Through these applications, LPI-LLM has the potential to become ICF-LLM, significantly advancing fusion research, paving the way for new insights and advancements in sustainable energy production.

**Reservoir Computing.** Traditional machine learning techniques (Sutton, 1988; Arel et al., 2010; Ahmed et al., 2010; Williams & Rasmussen, 2006; Samuel, 1959; Sapankevych & Sankar, 2009; Kadous, 1999) for scientific data often rely on transforming temporal inputs into high-dimensional state spaces using nonlinear mappings. RC introduced through key advancements like Echo State

Networks (Jaeger, 2007; Lukoševičius, 2012) and Liquid State Machines (Maass, 2011; Zhang et al., 2015), offers a compelling alternative to these traditional methods. RC operates by employing a fixed, untrained dynamic system—the reservoir—which processes input signals into rich representations. A notable advantage of RC is that only the readout layer is trained, drastically reducing computational overhead and simplifying the learning process. This efficiency makes RC particularly suited for large-scale scientific applications, where the volume and complexity of data demand scalable and low-cost solutions (Nakajima & Fischer, 2021; Schrauwen et al., 2007; Gauthier et al., 2021; Lukoševičius & Jaeger, 2009).

Our study proposes a paradigm shift by integrating LLMs into the RC framework, advancing beyond the limitations of traditional RNN-based reservoirs (Gauthier et al., 2021; Lukoševičius & Jaeger, 2009). Unlike standard reservoirs that may struggle with highly dynamic or noisy data, our method leverages the pre-trained knowledge and adaptive reasoning of LLMs, significantly enhancing RC's capacity to handle complex scientific datasets like `LPI`. This integration not only improves the system's ability to capture intricate temporal patterns but also minimizes the need for extensive parameter tuning. As a result, our approach offers a more powerful, adaptable, and efficient solution for scientific tasks, setting a new benchmark for RC-based methodologies.

**Time-series Forecasting.** In various scientific fields, time-series data plays a pivotal role, including `ICF`, where the temporal evolution of laser intensity significantly impacts target behavior and hot electron emission. Traditional methods for time-series analysis (Sutton, 1988; Arel et al., 2010; Williams & Rasmussen, 2006; Samuel, 1959; Sapankevych & Sankar, 2009; Kadous, 1999) typically involve the transformation of temporal inputs into high-dimensional spaces. However, these methods frequently encounter limitations when dealing with highly dynamic and complex data (Ahmed et al., 2010). Recently, transformer-based models for time-series forecasting, such as Temporal Fusion Transformers (TFT) (Lim et al., 2021), Informer (Zhou et al., 2021) and Autoformer (Wu et al., 2021), have demonstrated significant improvements by effectively capturing long-term dependencies and scaling to large temporal datasets. These models employ attention mechanisms that enhance accuracy, particularly in forecasting long-horizon data sequences. With the rise of large-scale models, the use of LLMs for time-series forecasting has emerged as a promising approach. For instance, Time-LLM (Jin et al., 2023) and GPT4TS (Zhou et al., 2023) have leveraged the vast pre-trained knowledge bases of LLMs to model complex temporal patterns more efficiently than previous methods. These LLM-based approaches benefit from their ability to handle varied, intricate time-series data with minimal task-specific fine-tuning, offering a flexible solution that adapts to diverse forecasting scenarios.

Conceptually different than these prior arts, our method introduces a novel architecture that synergistically combines multiple components to address the unique challenges of `LPI` forecasting in `ICF`. The Fusion-Specific Prompts strategically connect the LLM's vast knowledge base to `ICF`-specific nuances, enhancing the model's ability to interpret plasma physics phenomena. Our Signal-Digesting Channels, comprising temporal and spatial encoders, are specifically designed to capture the complex temporal patterns and critical landscape features of laser intensity signals in `ICF`. This multi-faceted approach allows LPI-LLM to more precisely model the intricate dynamics and uncertainties inherent in `LPI` data. By integrating these components, our framework achieves superior adaptability and robustness in time-series forecasting for `ICF` applications, overcoming limitations of previous methods in handling high-dimensional, volatile data scenarios typical in plasma physics.

## 5 CONCLUSION

Fusion energy stands as a pivotal pathway toward advancing human civilization to a Type I status on the Kardashev scale (Kardashev, 1964). The key to realizing this potential lies in mastering Inertial Confinement Fusion, where understanding laser-plasma instabilities is paramount. To address this challenge, we present LPI-LLM, a groundbreaking framework merging LLMs with reservoir computing. Our approach not only provides a cost-effective solution but also emerges as a top-tier contender in forecasting hot electron dynamics, offering invaluable insights for plasma scientists in refining `ICF` designs. Beyond its immediate impact on `ICF`, employing LLMs for scientific exploration holds promise for cross-domain applications, potentially catalyzing advancements in AI-driven scientific endeavors.

# 6 ETHICS

In our paper, which involves a new dataset, we will establish comprehensive ethical safeguards to mitigate potential misuse and ensure responsible utilization, as outlined in the detailed protocols in the final release of models and datasets. These protocols include strict usage guidelines, access restrictions, integration of safety filters, and monitoring mechanisms. We conduct thorough risk assessments to identify potential misuse scenarios, developing tailored mitigation strategies such as robust data governance frameworks. Although not all research may require stringent safeguards, we adhere to best practices, promoting ethical awareness encouraging researchers to consider the broader impacts of their work and maintain detailed documentation for transparency and accountability. These efforts demonstrate our commitment to upholding the highest standards of ethical conduct in scientific inquiry, aiming to safeguard the interests and privacy of all people involved.

# 7 REPRODUCIBILITY

LPI-LLM is implemented in PyTorch (Paszke et al., 2019). Experiments are conducted on two NVIDIA A100-40GB GPUs. To guarantee reproducibility, we fully describe our approach in §2 and implementation details in Appendix §S2. Our full implementation of code, model weights, and `test` split of the dataset are also submitted with this paper for reproduction, which can be accessed via the anonymous link: https://anonymous.4open.science/r/LPI-LLM.

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

- §S1 contains **Glossary**.
- §S2 provides **Implementation Details**.
- §S3 reports more **Quantitative Results** with **Runtime Analysis**.
- §S4 shows more **Qualitative Results**.
- §S5 summarizes **Impact of Prompt Descriptors**
- §S6 analyzes **Failure Case**.
- §S7 conducts **Confidence Analysis**.
- §S8 discusses the **Social Impact & Limitation** of our research.
- §S9 supplies **Data License** for the methods we used for comparison.

## S1  GLOSSARY

Given that this paper presents our work in the field of AI for Physics, it inevitably involves numerous specialized terms originating from both physics and AI. To ensure that audiences from both domains can benefit from our work, we have compiled key terms from each field, along with their definitions, to facilitate readers' understanding.

- **Attention Mechanism**: A technique in neural networks that allows the model to focus on different parts of the input data with varying levels of importance when making predictions, improving accuracy for tasks like translation and text summarization.
- **Cross-Beam Energy Transfer (CBET)**: Exchange of energy between intersecting laser beams in plasma, driven by scattering processes. It affects energy delivery in inertial confinement fusion, influencing implosion efficiency and uniformity.
- **Embedding**: A way to represent data like words or signals as numerical vectors in continuous space, capturing relationships and enabling analysis.
- **Entropy**: A measure of uncertainty in a model's predictions, with high entropy indicating more randomness and low entropy indicating greater confidence.
- **Few-shot Learning**: The ability of AI models to perform tasks using only a small number of examples, allowing efficient learning in situations with limited data.
- **Fine-tuning**: A process of adapting a pre-trained model to a specific task or dataset by continuing its training, improving performance on specialized applications.
- **Hard X-Rays (HXR)**: High-energy X-ray radiation emitted when hot electrons interact with plasma. It is the primary diagnostics for hot electrons.
- **Hot Electrons**: Energetic electrons generated during laser-plasma interactions that can degrade fusion performance.
- **Inertial Confinement Fusion (ICF)**: A type of fusion energy research where fuel targets are compressed by intense drivers to achieve fusion conditions.
- **Large Language Models (LLMs)**: AI systems trained on vast datasets to understand and generate human-like content, excelling in tasks such as answering questions and reasoning.
- **Laser-Plasma Instabilities (LPI)**: Physical phenomena that occur when intense laser light interacts with plasma, potentially disrupting the fusion process.
- **Neural Network**: A computing system inspired by biological brains, consisting of interconnected nodes organized in layers that can learn patterns from data through training.
- **Particle-In-Cell (PIC) Simulation**: A first-principle computational method for simulating plasma physics by tracking particles and fields.

- **Reservoir Computing**: A framework that processes input through a fixed, random network (reservoir) to generate high-dimensional representations, useful for analyzing time-series and dynamic data.
- **Token**: A fundamental unit of text or data processed by a model, such as a word, subword, or character, used in tasks like text generation or analysis.
- **Transformer**: A neural network architecture that uses self-attention mechanisms to process sequential data, enabling efficient handling of context and relationships in sequences.
- **Two-Plasmon Decay (TPD)**: A plasma instability in which an electromagnetic wave splits into two plasma waves, leading to the production of hot electrons. These electrons can preheat the target, elevate the shell entropy, and reduce the efficiency of the implosion.

## S2 IMPLEMENTATION DETAILS

The overall pipeline of LPI-LLM is shown in Fig. 3. Experiments are conducted on two NVIDIA A100-40GB GPUs. For our approach, we keep all parameters of the LLMs and most of the SDC frozen during the fine-tuning. Only parameters pertaining to the Prediction Head and partial Spatial Encoder are trainable. The codes and dataset shall be publicly released upon paper acceptance.

- LPI-LLM is built from `Llama 2 7B` (Touvron et al., 2023a), `Llama 3 8B` (AI, 2024), `Gemma 2 9B` (Team, 2024) and `OLMo` (Groeneveld et al., 2024) to construct reservoir without tuning.
- *Fusion-specific prompts* structure the textual prompts with three descriptors: context descriptor, task descriptor, and input descriptor. Each descriptor is initialized with specialized tokens for indication (*e.g.*, $<|begin\_of\_text|>$, $<|eot\_id|>$, $<|start\_header\_id|>$, *etc*) and input scalars as context descriptions (*e.g.*, $<seq\_len>$, $<pred\_len>$, $<phase\_plate>$, *etc*). These prompts are subsequently concatenated and input into the projection layer from LLMs for feature embedding.
- *Signal-digesting channels* are composed of two components, temporal encoder and spatial encoder. The former one, which incorporates 24 Transformer layers and a linear layer, captures temporal features over the input laser signal. This module has been pre-trained on the Large-scale Open Time Series Archive (LOTSA) dataset (Woo et al., 2024), which covers nine varied domains and compiles over 27 billion timestamped instances. The spatial decoder first uses a projection block from LLMs to encode the context description of the input signal, followed by a leaner transformation. Outputs are fed into a cross-attention layer, where Key and Value are derived from the contextual embedding and query stems from temporal features, to generate the final spatial features. We concatenate the spatial and temporal features before feeding them into a linear layer to produce the final, augmented input signals.
- *Confidence scanner* has been described in §2.2.3 and it has no consumption of parameters. The default number of tokens $k$ used in confidence calculation is set to 50 in the implementation.
- *Prediction head* consists of two layers: a convolution layer with the kernel size of 32 and stride of 32, followed by batch normalization and `GELU` activation, connected to LLM, then fed to a linear layer with the input dimension of 128 that produces the final prediction.

## S3 QUANTITATIVE RESULTS

This section elaborates on a detailed analysis of quantitative results in Table S1, focusing specifically on in-context learning (Brown et al., 2020) performance and a runtime assessment of the models under investigation. Initially, we present supplementary in-context learning results obtained directly from various LLMs (*i.e.*, Llama 2 (Touvron et al., 2023b), Llama 3 (AI, 2024), and Claude 3 Opus (Anthropic, 2024)). These findings indicate that, even without an additional fine-tuning process, the LLMs exhibit substantial proficiency in the in-context learning scheme within the ICF task. For instance, Claude 3 Opus (Anthropic, 2024) achieves CAE scores of 12.19, 10.67, and 9.46 for the 1-shot, 2-shot, and 3-shot scenarios, respectively. It is amply demonstrated that the vanilla LLM has the ability to make inferences and predictions on empirical scientific data even if it is not fine-tuned at all. This underscores that our approach, leveraging these LLMs, represents a notable advancement, particularly in forecasting the energy dynamics of hot electrons.

Furthermore, it is pertinent to emphasize the economic and operational advantages of our computational approach over traditional physical experiments. Specifically, conducting a single ICF

Table S1: **Quantitative results** on LPI4AI `test` split for hot electron energy forecasting (see §3.1 for details). Train Time refers to training for the designated task, and Infer Time refers to the amount of time used to predict one case. Note that 3-shot experiments could not be performed on `Llama` series of models due to the limitation of the context window.

| Method | # Params | Train Time | Infer Time | CAE↓ | top-1 MAE↓ | top-5 MAE↓ |
|---|---|---|---|---|---|---|
| PIC Simulation | - | - | > 10 hrs | 2.88 | 0.20 | 0.13 |
| LSTM | 81.6K | ∼5 mins | < 1 s | 5.82 | 0.35 | 0.35 |
| Autoformer | 120.4K | ∼8 mins | < 1 s | 5.79 | 0.35 | 0.34 |
| GPT4TS | 1.5 B | ∼22 mins | ∼2 s | 3.34 | 0.18 | 0.14 |
| Time-LLM | 7 B | ∼20 mins | ∼3 s | 3.48 | 0.18 | 0.15 |
| Llama 2 (1-Shot) | 7 B | - | ∼3 s | 471.22 | 15.80 | 14.92 |
| Llama 2 (2-Shot) | 7 B | - | ∼5 s | 30.33 | 0.95 | 0.91 |
| Llama 2 (1-Shot) | 70 B | - | ∼7 s | 20.63 | 0.64 | 0.63 |
| Llama 2 (2-Shot) | 70 B | - | ∼8 s | 16.42 | 0.51 | 0.50 |
| Llama 3 (1-Shot) | 8 B | - | ∼4 s | 583.14 | 17.70 | 16.97 |
| Llama 3 (2-Shot) | 8 B | - | ∼6 s | 26.66 | 0.83 | 0.81 |
| Llama 3 (1-Shot) | 70 B | - | ∼14 s | 72.35 | 1.02 | 1.15 |
| Llama 3 (2-Shot) | 70 B | - | ∼19 s | 13.62 | 0.40 | 0.39 |
| Claude 3 Opus (1-Shot) | 137 B | - | ∼12 s | 12.19 | 0.39 | 0.39 |
| Claude 3 Opus (2-Shot) | 137 B | - | ∼17 s | 10.67 | 0.38 | 0.37 |
| Claude 3 Opus (3-Shot) | 137 B | - | ∼20 s | 9.46 | 0.37 | 0.36 |
| RCRK | 106 K | ∼2 mins | < 1 s | 4.31 | 0.28 | 0.22 |
| HoGRC | 394 K | ∼4 mins | < 1 s | 4.20 | 0.25 | 0.22 |
| NGRC | 157 K | ∼2 mins | < 1 s | 4.28 | 0.27 | 0.23 |
| **LPI-LLM (Gemma-2)** | 9 B | ∼30 mins | ∼4 s | 2.04 | 0.14 | 0.12 |
| **LPI-LLM (OLMo)** | 7 B | ∼30 mins | ∼3 s | 1.97 | 0.14 | 0.12 |
| **LPI-LLM (Llama-2)** | 7 B | ∼30 mins | ∼3 s | 2.15 | 0.14 | 0.12 |
| **LPI-LLM (Llama-3)** | 8 B | ∼30 mins | ∼4 s | 1.90 | 0.14 | 0.11 |

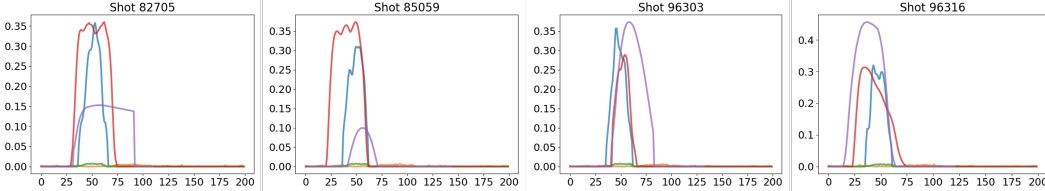

Figure S1: Predictions of **LLMs with In-Context Learning**. We plot Ground Truth and the predictions of Claude 3 Opus (3-shot) and Llama 3 70B (2-shot) with the comparison of trained methods LSTM and Autoformer. Y and X axes denote energy and time steps.

experiment typically incurs costs upwards of one million US dollars. Conversely, computational simulations such as a $150\mu m$ PIC simulation (Cao et al., 2022) require extensive computational resources, amounting to the utilization of 19,584 cores of CPU over a period of 10 hours. In stark contrast, our model necessitates significantly less computational time and resources, requiring only 30 minutes on 2 NVIDIA A100 GPUs for training and only 3∼4 seconds for inference with much higher predictive accuracy compared to the PIC simulation. This comparison not only underscores the cost-effectiveness of our approach but also its efficiency and practicality in other scientific applications where computational resource constraints are a critical factor.

## S4 MORE QUALITATIVE RESULT

This section expands to include more qualitative results that help to understand the capabilities and effectiveness of this model. Initially, we release all visualized prediction results of our model LPI-LLM on `test` split of our dataset LPI4AI in Fig. S2. From these qualitative results, it can be found that our model achieves accurate predictions on all unseen data, especially conforming to the

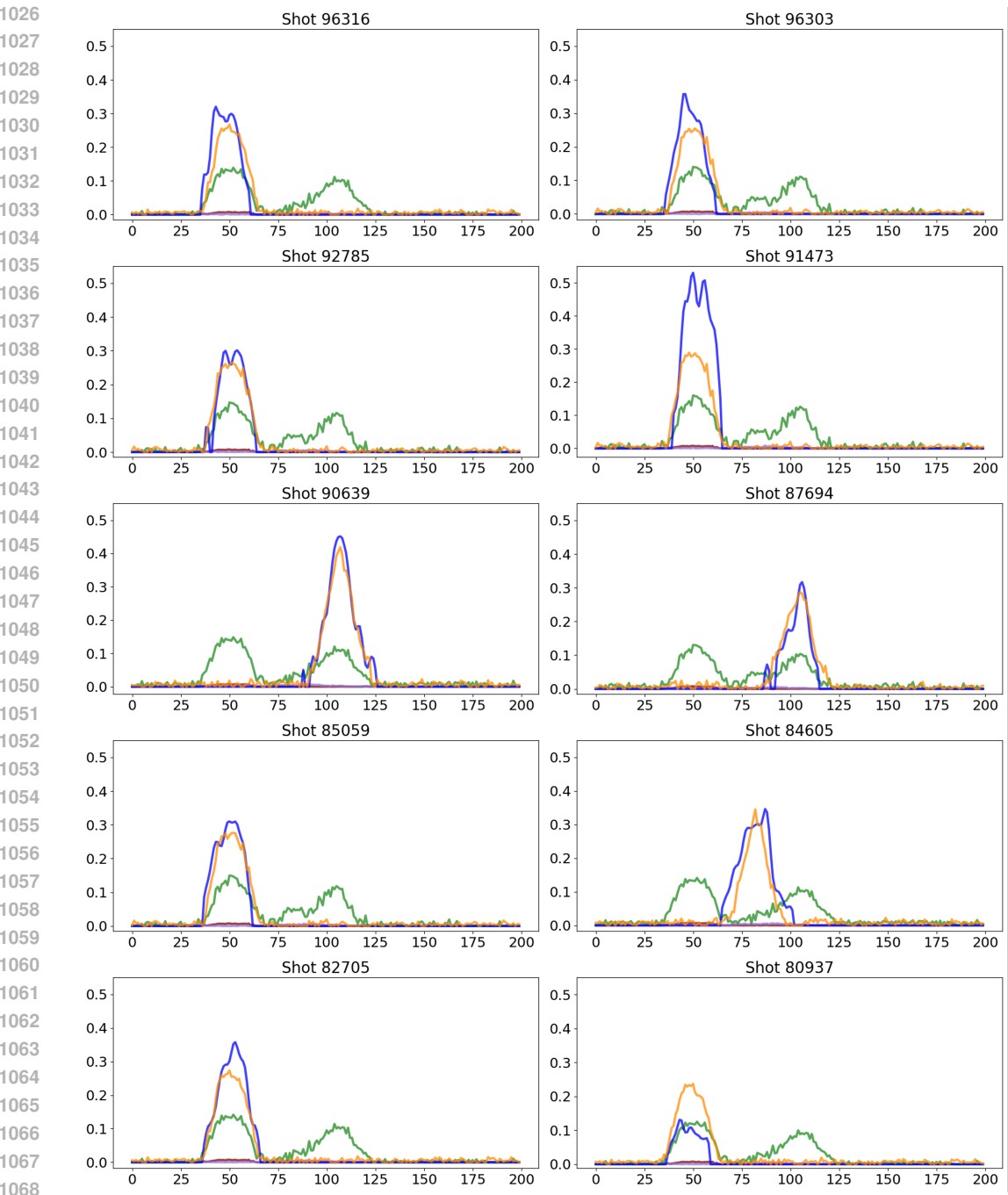

Figure S2: Visualization of **hot electron prediction** in `test` split. We plot Ground Truth and the predictions of Ours, Time-LLM, LSTM and Autoformer. Y and X axes denote energy and time steps, respectively.

temporal and spatial characteristics of the predicted targets, which is crucial for physicists to apply our model as a tool in the design of real-world `ICF` shots.

Recall that we describe in §S3, the direct prediction of `ICF` tasks by vanilla LLM use in-context learning method without fine-tuning on our data is not as good as fine-tuned methods in terms of the quantitative value of metric, but it can in fact predict more meaningful patterns than traditional methods such as LSTM (Hochreiter & Schmidhuber, 1997) and Autoformer (Wu et al., 2021). As

illustrated in Figure S1, the LLM without fine-tune can infer approximate predictions with reference to the 1 to 3 examples provided, as compared to LSTM (Hochreiter & Schmidhuber, 1997) and Autoformer (Wu et al., 2021) which can only predict straight lines close to 0 with patterns. This proves that the vanilla LLMs themselves already contain the capability to infer specific empirical scientific data, which is the core reason we chose LLMs as the reservoir of our LPI-LLM.

## S5 IMPACT OF PROMPT DESCRIPTORS

Table S2: A full set of ablative studies of Fusion-specific Prompt on `val` split.

| Descriptors | CAE↓ |
|---|---|
| BASELINE | 2.01 |
| w/ Context Descriptor | 1.52 |
| w/ Task Descriptor | 1.49 |
| w/ Input Descriptor | 1.58 |
| w/ Context + Task Descriptors | 1.46 |
| w/ Context + Input Descriptors | 1.34 |
| w/ Task + Input Descriptors | 1.33 |
| **OURS** (**all three**) | 1.19 |

To assess the effects of individual and combined prompt descriptors (context, task, and input), futher experiments in addtion to those present in Table 2b were conducted to evaluate performance with each individual descriptors and pairwise combinations. As present in Table S2, the results showed that all individual descriptors significantly improved baseline from 2.01 in terms of CAE. The task descriptor achieved the best individual performance with a CAE of 1.49, closely followed by the context descriptor at 1.52. The input descriptor, though slightly less impactful alone, still provided notable improvement with a CAE of 1.58.

Pairwise combinations demonstrated synergistic effects, with context + input and task + input achieving CAE scores of 1.34 and 1.33, respectively, outperforming single-descriptor setups. The combination of context + task yielded a CAE of 1.46, showing balanced improvement. The full integration of all three descriptors (context, task, and input) resulted in the best performance, with a CAE of 1.19. These findings highlight the complementary nature of the descriptors, with the input descriptor playing a crucial role when paired with others, and the fusion-specific prompt design proving essential for optimal system performance.

## S6 FAILURE CASE ANALYSIS

In this section, we examine a significant outlier with the largest error forecasts generated by the LPI-LLM on the `test` split. This particular instance serves as a critical case study for understanding the limitations and challenges faced by our model. Fig. S3 illustrates that the shot markedly deviates from the typical scenarios. Notably, this shot exhibits an exceptionally low peak hot electron energy, registering less than 0.15, whereas the majority of other cases yield values ranging between 0.25 and 0.5 under a comparable input laser profile. This anomaly categorizes this shot as an out-of-distribution (OOD) instance. The limited volume of training data available for LPI-LLM is a plausible explanation for the model's diminished performance on this OOD data. In scenarios where training data is sparse, the model's capability to generalize to new, especially atypical, data points is inherently restricted. Consequently,

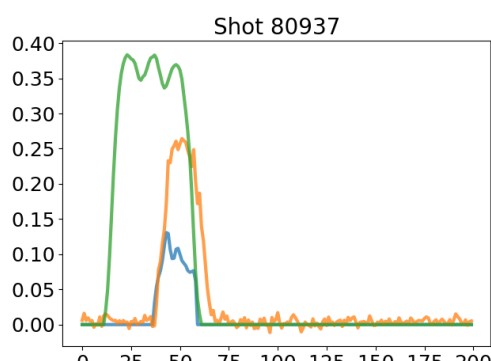

Figure S3: The Qualitative result of Shot 80937. We plot Input Laser Intensity, Ground Truth and Prediction. Y axis denotes laser intensity / hot electron energy, and X axis denotes time step.

this case highlights the importance of enhancing the dataset's diversity and volume for `ICF` tasks. We hope the community can share more data points to improve and enlarge LPI4AI dataset (§3.1) together.

## S7 CONFIDENCE ANALYSIS

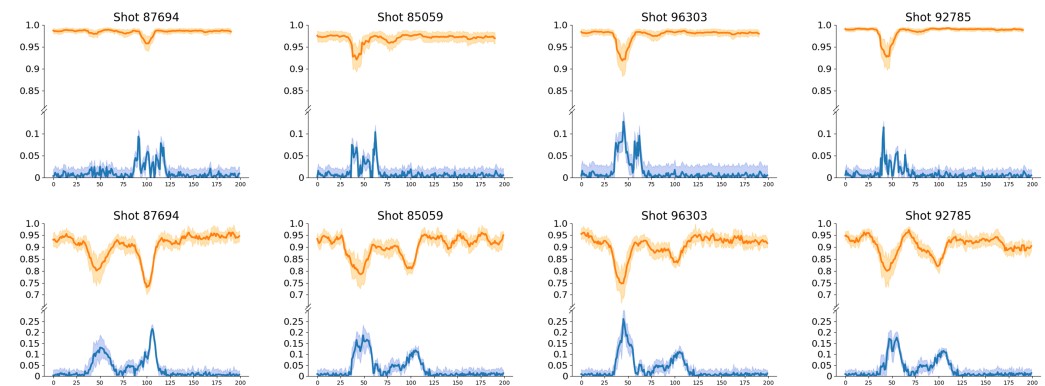

Figure S4: Qualitative results of **confidence score** and **prediction error**. Comparison between our LPI-LLM (first row) and Time-LLM (second row).

In this section, we provide a comprehensive discussion and analysis of the confidence scores associated with the LPI-LLM. As elaborated in §2.2.3, our confidence scores offer per-step evaluations, thereby aiding physicists to gain deeper insights into the reliability across various segments of predictions. This functionality is particularly vital for understanding the model's performance dynamics within specific contexts of its predictive output.

To visually represent the relationship between prediction errors and confidence scores, Figure S4 compares the performance of our LPI-LLM with that of the baseline Time-LLM model. The figure shows the prediction errors and corresponding confidence scores for four test sets, revealing

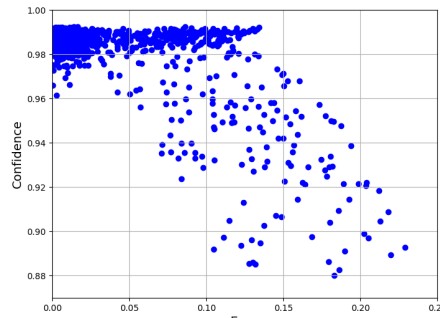

Figure S5: Qualitative results of correlation between confidence scores and errors of our model's prediction.

a clear pattern: confidence scores decline noticeably during intervals where errors are higher. For LPI-LLM, this decline is smoother and more localized, reflecting a consistent ability to adapt to challenging phases. In contrast, the baseline Time-LLM exhibits a more volatile confidence profile, with sharper fluctuations and a pronounced drop during critical peak energy periods. This suggests that Time-LLM struggles to maintain reliable predictions during high-stakes moments, whereas LPI-LLM demonstrates greater stability. This difference underscores several advantages of our approach. By integrating domain-specific enhancements, such as a tailored prompt structure and signal-digesting channels designed to handle the temporal and spatial complexities of ICF data, LPI-LLM maintains a more robust performance under pressure. In comparison, the baseline lacks these refinements, resulting in less stable confidence levels and weaker reliability when precise predictions matter most. Overall, Figure S4 highlights the value of our model's confidence scores not just as uncertainty estimates, but as a diagnostic tool for pinpointing intervals where predictions may require closer scrutiny—an essential capability for high-stakes predictive tasks.

Building on this analysis, Figure S5 quantifies the observed relationship between confidence scores and prediction errors, demonstrating a clear negative correlation across the `test` split. Lower confidence scores consistently coincide with higher prediction errors, validating the reliability of the LPI-LLM's confidence scores as uncertainty estimates for the ICF task. This underscores the importance of confidence scores as a diagnostic tool, highlighting intervals where the model's predictions are potentially less reliable. By mapping these confidence scores to the corresponding prediction errors, physicists can identify specific phases within the prediction and temporal sequence where the model's forecasting should be interpreted with caution. This capability not only enhances the trustworthiness of the LPI-LLM but also provides critical feedback for further refinement of the model in the future research.

Moreover, the integration of confidence scores into the model's predictive framework offers a robust mechanism for assessing the model's performance in real-time applications. By continuously mon-

itoring these scores, physicists can make informed decisions about the reliability of the predictions, ensuring that critical assessments and subsequent actions are based on the most credible forecasting.

## S8  SOCIAL IMPACTS AND LIMITATIONS

The introduction of LPI-LLM represents a significant advancement in integrating LLMs with classical reservoir computing paradigms to enhance predictive capabilities in Inertial Confinement Fusion. This novel approach not only meets but exceeds several existing state-of-the-art models in performance benchmarks. From a societal perspective, the implications of LPI-LLM are profoundly beneficial, as our approach provides a valuable tool for advancing our understanding and capabilities in harnessing fusion energy — a potential key to long-term sustainable energy solutions. However, it is imperative to acknowledge and critically assess the potential drawbacks associated with this technology. Similar to other predictive models, LPI-LLM faces challenges when dealing with out-of-distribution data or scenarios that have not been previously encountered. This limitation underscores the need for ongoing research and refinement, particularly in its application to real-world `ICF` scenarios where unpredictable behaviors might emerge. Therefore, while the model demonstrates promising applications, its deployment in practical settings must be approached with caution, ensuring continuous evaluation and adaptation to maintain reliability and safety in its prediction.

## S9  LICENSES FOR EXISTING ASSETS

All the methods we used for comparison are publicly available for academic usage. PIC Simulation is implemented based on the reproducing by `osiris-code/osiris` with AGPL-3.0 license. We use `huggingface/transformers` for the implementations of Autoformer (Wu et al., 2021) under Apache-2.0, Llama 2 (Touvron et al., 2023b) under Llama 2 Community License, Llama 3 (AI, 2024) under Llama 3 Community License, Gemma 2 (Team, 2024) under Gemma Terms of Use and OLMo (Groeneveld et al., 2024) under Apache-2.0. We used the official repositories `DAMO-DI-ML/NeurIPS2023-One-Fits-All` (GPT4TS (Zhou et al., 2023)), `KimMeen/Time-LLM` (Jin et al., 2023), `rubenohana/Reservoir-computing-kernels` (RCRK (Dong et al., 2020)), `CsnowyLstar/HoGRC` (Li et al., 2024) and `quantinfo/ng-rc-paper-code` (NGRC (Gauthier et al., 2021)) for our comparison experiments, where Time-LLM (Jin et al., 2023) is licensed under Apache-2.0, HoGRC (Li et al., 2024) and NGRC (Gauthier et al., 2021) are licensed under MIT.

