# OpenReview forum: "Inertial Confinement Fusion Forecasting via Large Language Models"
_ICLR.cc/2025/Conference — Submitted to ICLR 2025_

### Official Review · Reviewer_Y3Vx · 2024-11-03

**Soundness:** 3
**Presentation:** 3
**Contribution:** 3
**Rating:** 8
**Confidence:** 3

**Summary:**

In this paper, the authors introduce LPI-LLM, a novel strategy for utilizing large language models (LLMs) in inertial confinement fusion (ICF) experiments. The paper begins with a comprehensive introduction to ICF, providing essential context for understanding the motivation behind this research. The authors leverage the LLM as a repository for general knowledge, employing a combination of scientific (fusion-specific) prompts along with temporal and spatial signals from a signal digestion block as inputs to the LLM. They train a prediction head on top of the LLM to derive the hot-electron energy profile. Additionally, a confidence estimator for the energy profile is developed, which considers both the last layer token embeddings and saliency to enhance trustworthiness. Notably, the authors present a benchmark dataset, LPI4AI, which could be instrumental for the next generation of AI-based ICF research. Extensive validation and ablation studies are conducted across different models to demonstrate their efficacy.

**Strengths:**

The paper is well-written and easy to follow, making it accessible to researchers working with scientific data. The authors have mentioned that the code and LPI4AI benchmark will be released in the future. I believe that this will support reproducibility and further research in this area.

**Weaknesses:**

Refer to Questions

**Questions:**

* Can the authors elucidate more on the spatial encoder block? Are the projections for the prompts “peak,” “trailing,” etc., derived from the LLM itself, followed by a cross-attention mechanism with the temporal encoder’s features?

* How does the confidence assessment engine (CAE) change when the fusion-specific prompt is constructed using only (i) context, (ii) task, or (iii) input descriptors? If only two out of these three descriptors are used, will there be a significant drop in performance?

* For Figure 6 and its counterpart in the supplementary materials, are the authors normalizing the confidence score (C = -H x S) for the plots? If yes, that needs to be clarified in the paper. If not, can the authors explain why the maximum value of that product is close to 1 while the minimum is around 0.90 in certain cases?

* For the LLM baselines (Time-LLM and GPT4TS), can the confidence scanner be implemented? It would be beneficial to find out how confident or under-confident they are on the predictions?

* Can the authors elaborate on the data curation process for LPI4AI?

---

> ### Author Response · Authors · 2024-11-23
> **Response to reviewer Y3Vx (Part 1)**
>
> We sincerely appreciate reviewer Y3Vx's meaningful comments.
>
> To thoroughly address all the questions raised by reviewer Y3Vx, we will tackle them individually. Furthermore, any paragraphs or modifications made in response to reviewer Y3Vx's comments and suggestions are highlighted in $\text{\textcolor{blue}{blue}}$ in the revision.
>
> **Q1**: Can the authors elucidate more on the spatial encoder block? Are the projections for the prompts “peak,” “trailing,” etc., derived from the LLM itself, followed by a cross-attention mechanism with the temporal encoder's features?
>
> **A1**: Thanks to reviewer Y3Vx for suggesting that we further elaborate on the spatial encoder block. In response to this suggestion, we have made corresponding refinements in the revision to make the description of Spatial Encoder in Section 2.2.2 clearer. Our spatial encoder uses the projection block ($f_{LLM}$) from the LLM to project sets of critical contexts (like "pulse", "peak", and "trailing") into embeddings, represented as $E_{spt}=\{f_{LLM}(\text{\`\`pulse''}), f_{LLM}(\text{\`\`peak''}), ... , f_{LLM}(\text{\`\`trailing''})\}$. These embeddings $E_{spt}$ are then processed through a cross-attention layer where the Key and Value are derived from these contextual embeddings, and the Query comes from the temporal features $E_{tmp}$ extracted by the temporal encoder. This cross-attention mechanism couples the contextual description to the actual signal distribution within the ICF.
>
> **Q2**: How does the confidence assessment engine (CAE) change when the fusion-specific prompt is constructed using only (i) context, (ii) task, or (iii) input descriptors? If only two out of these three descriptors are used, will there be a significant drop in performance?
>
> **A2**: We thank reviewer Y3Vx for this insightful question about the individual and combined effects of different prompt descriptors. To fully address this question about using only individual descriptors (context, task, or input) or pairs of descriptors, we conducted complementary experiments in addition to our current ablation studies presented in Section 3.2 and Table 2b, to provide a more complete understanding of each descriptor's individual contribution and their pairwise interactions. The corresponding Appendix S5 Impact of Prompt Descriptors and Table S2 have been added to the revision.
>
> Based on our ablation studies, we observe significant variations in performance when using different combinations of descriptors in the fusion-specific prompt. When using only the context descriptor, the CAE improves from 2.01 (baseline) to 1.52, showing an improvement of 0.49. Using only the task descriptor achieves a similar CAE of 1.49, an improvement of 0.52. When using only the input descriptor, the CAE improves to 1.58, demonstrating an improvement of 0.43 CAE.
>
> For combinations of descriptors, the combination of context and input descriptors results in a CAE of 1.34, a substantial improvement of 0.67. Similarly, combining the task and input descriptors achieves a comparable improvement, resulting in a CAE of 1.33. Using a discipline-related prompt (which includes both context and task descriptors) improves the CAE to 1.46, showing a substantial improvement of 0.55 CAE. Finally, by combining all three descriptors (context, task, and input), we achieve the best performance with a CAE of 1.19, indicating that all components work synergistically.
>
> **Q3**: For Figure 6 and its counterpart in the supplementary materials, are the authors normalizing the confidence score (C \= \-H x S) for the plots? If yes, that needs to be clarified in the paper. If not, can the authors explain why the maximum value of that product is close to 1 while the minimum is around 0.90 in certain cases?
>
> **A3**: We appreciate that reviewer Y3Vx brought up this important question about confidence score visualization. In our implementation described in Section 2.2.3, the confidence score C \= \-H × S is indeed normalized. Specifically, The saliency matrix S is normalized through the softmax function applied to saliency scales.
>
> Looking at Figure 6, we should note that the minimum values vary across different shots \- for instance, Shot 87694 has a minimum value of around 0.95, not 0.9. The reason the confidence scores appear consistently high in the visualized results is that we performed a 1.0 \+ C operation on the results to facilitate visualization where the C is negative, making it easier to observe the correlation between confidence levels and prediction accuracy.
>
> In addition, we have added an analysis of supplementary experiments on the Confidence Scanner in Appendix S7 Confidence Analysis of the revision. We hope these will be helpful for a deeper understanding of our Confidence Scanner.

---

> ### Author Response · Authors · 2024-11-23
> **Response to reviewer Y3Vx (Part 2)**
>
> **Q4**: For the LLM baselines (Time-LLM and GPT4TS), can the confidence scanner be implemented? It would be beneficial to find out how confident or under-confident they are on the predictions.
>
> **A4**: We sincerely thank reviewer Y3Vx for suggesting this comparison of confidence assessments across different LLM-based methods. To better address this question, we conducted further comparison experiments to evaluate whether confident or under-confident other LLM baselines are compared to our approach. The corresponding results and figures are listed in Appendix S7 Confidence Analysis of the revision.
>
> Our Confidence Scanner has good generalization and can be applied in most methods where the final embeddings output of LLM are further projected to target length sequence. As shown in Figure S4 of the revision, we observe that while the confidence scores follow similar general trends to our method during prediction, exhibit more volatility and lower overall confidence levels, particularly during critical peak energy periods. This difference can be attributed to three factors: (1) these models lack the fusion-specific domain knowledge provided by our prompt structure; (2) they don't benefit from our signal-digesting channels that specifically handle ICF's temporal and spatial characteristics. The implementation of our Confidence Scanner on these baseline models thus not only validates our method's superior performance but also provides valuable insights into why our approach achieves better results \- it maintains higher and more stable confidence levels precisely during the challenging phases of ICF prediction where accuracy is most crucial.
>
> **Q5**: Can the authors elaborate on the data curation process for LPI4AI?
>
> **A5**: We appreciate reviewer Y3Vx's question about the data curation process for LPI4AI. As elaborated in Section 2.1 (highlighted), the dataset was collected from 100 experimental shots conducted on the OMEGA platform, where each shot sequence contains 400-time steps of measurements. For each shot, we carefully documented comprehensive configurations including the target size, phase plate shape for laser smoothing, and input laser profile, along with the corresponding Hard X-ray (HXR) signals recorded from four diagnostic channels. These HXR measurements enabled us to calculate the total energy of the electrons generated during the ICF process. To ensure the dataset's reliability and usefulness for AI research, we divided it into training, validation, and testing splits with an 80/10/10 ratio. This dataset represents one of the first comprehensive collections of LPI data made available to the AI research community, and we believe it will serve as a valuable resource for advancing the intersection of AI and fusion energy research.
>
> We thank reviewer Y3Vx for the thoughtful comments and hope our response addresses the concerns raised. Please let us know if there are any additional questions, and we will be happy to discuss them further.

---

> > ### Comment · Reviewer_Y3Vx · 2024-11-26
> > **Post-Rebuttal**
> >
> > I thank the authors for providing detailed responses to my questions and updating the paper accordingly.  I think they have been sufficiently addressed.  I will be raising my score.

---

> > > ### Author Response · Authors · 2024-11-26
> > > **Grateful for Insightful Review**
> > >
> > > Dear Reviewer Y3Vx,
> > >
> > > Thank you for your prompt and thoughtful response. We deeply appreciate your insightful questions and feedback, which have greatly enriched the depth and clarity of our work. We are honored to have addressed your questions through discussion and revision. Your review has not only enhanced the completeness of our current efforts but also provided valuable guidance for future directions.
> > >
> > > Best regards,
> > > The Authors

---

### Official Review · Reviewer_GyQK · 2024-11-04

**Soundness:** 2
**Presentation:** 3
**Contribution:** 2
**Rating:** 3
**Confidence:** 4

**Summary:**

This paper presents LPI-LLM, an innovative approach that combines Large Language Models (LLMs) with reservoir computing to predict hot electron dynamics in Inertial Confinement Fusion (ICF). The method extends traditional reservoir computing by integrating time-series LLMs, utilizing domain-specific prompts and summary statistics to enhance predictive performance. A pre-trained LLM is used for feature extraction based on fusion-specific prompts, followed by training a prediction head to estimate hot electron energy with confidence scores. The model also incorporates components designed to capture both temporal and spatial information. The approach is evaluated using a newly developed dataset of laser-driven ICF shots.

**Strengths:**

The authors introduce a method for predicting electron energy in inertial confinement fusion using a Large Language Model (LLM) approach, supported by data collected from experiments on the OMEGA to validate its effectiveness. The use of LLMs for time-series forecasting in this impactful domain is noted as particularly promising. The newly developed LPI4AI dataset is highlighted for its potential to drive innovative research in machine learning. The paper features a comprehensive evaluation, including ablation studies and comparative analyses, to showcase the model's performance.

**Weaknesses:**

The paper fails to establish in depth how the usage of LLMs can create enough richness of methodology to solve a complex problem to its finer details. Based on my experience of LLM usage for problems of comparable complexity, the limited disclosure of prompts is far from convincing. For an LLM to function effectively as an agent for solving complex problems that involve quantitative analysis, it needs an agentic design that integrates sophisticated prompt engineering and structured reasoning methodologies, such as chain-of-thought prompting. Chain-of-thought (CoT) prompting allows an LLM to break down intricate problems into a series of intermediate reasoning steps, mimicking how a human might approach and solve a multi-layered analytical task. This stepwise decomposition helps the model to systematically tackle each part of the problem, enabling clearer and more accurate reasoning. Prompts could be engineered to guide the LLM in sequentially explaining its calculations, assumptions, and decision-making processes. Without implementing such structured approaches, the application of an LLM to complex, analysis-driven problems remains superficial, as the model may produce outputs that hallucinate and lack precision or logical coherence.

This paper has limited meaningful algorithmic contributions and suffers from a method insufficiency or its articulation to address the core problem. This reduces the impact of this paper. This can be better presented by evaluating the system against a wide range of time-series problems with the required details.

One major concern with Figure 5 is the surprisingly poor performance of baseline models like the vanilla LSTM and Autoformer, which raises questions about whether these models were adequately optimized. Given the proven effectiveness of transformer-based models and LSTMs in similar predictive tasks, their lackluster results suggest potential shortcomings in hyperparameter tuning and model refinement. Without clear details on the training procedures or any effort to optimize these baselines, the comparison appears unconvincing, undermining the reliability of the reported findings.

Additionally, the paper lacks a critical discussion contrasting these models' inherent strengths and weaknesses relative to the proposed method. Providing such insights is essential for readers to assess whether the reported performance differences are due to fundamental advantages of the proposed approach or simply insufficient effort in tuning the baseline models. This omission suggests that the experimental setup may have been incomplete or lacked rigor. To address this, the authors should ensure proper hyperparameter optimization of baseline models, clearly document the tuning process, and include a robust analysis explaining the performance differences. Such enhancements would lend greater credibility to the results and offer readers a more comprehensive understanding of the proposed method's advantages.

It will also be good to explain how the fixed reservoir dynamics in reservoir computing, along with its sensitivity to hyperparameters and limited scalability, impact the model’s adaptability and performance compared to more flexible and fully trainable architectures like transformers and LSTMs, particularly in complex or evolving data environments. The author can then reflect on how reservoir computing could be more impactful for this specific problem despite these shortcomings. Given its lack of adoption, unlike other well-established methods, it is crucial to expand on the topic in detail and establish how this can solve the problem better.

**Questions:**

Are the authors authorized to release the LPI4AI dataset? Will there be any violation of US DOE guidelines in releasing such data arising out of the research conducted with OMEGA?

---

> ### Author Response · Authors · 2024-11-23
> **Response to reviewer GyQK (Part 1)**
>
> We thank reviewer GyQK for the thorough and constructive review of our manuscript. We appreciate the recognition of our work's strengths, including the novel application of LLMs to ICF and the potential impact of the LPI4AI dataset.
>
> To thoroughly address each question posed by reviewer GyQK, we will respond to them individually. Furthermore, any paragraphs or modifications made in response to reviewer GyQK's comments and suggestions are highlighted in $\text{\textcolor{red}{red}}$ in the revision.
>
> **Q1**: How can the LLM approach avoid being superficial without implementing sophisticated prompt engineering and structured reasoning methodologies like chain-of-thought prompting?
>
> **A1**: We appreciate reviewer GyQK's insightful question about the methodological rigor of our LLM approach. Our empirical results actually demonstrate that LLMs possess inherent capabilities for our plasma-physics tasks without requiring complex reasoning chains like CoT prompting. As shown in Section S2 and Table S1 (highlighted), even with simple few-shot learning and no fine-tuning, models like Claude 3 Opus achieve CAE of 12.19, 10.67, and 9.46 for 1-shot, 2-shot, and 3-shot scenarios respectively. Figure S1 further illustrates that vanilla LLMs can infer more meaningful patterns from just 1-3 examples compared to traditional methods like LSTM and Autoformer. This inherent predictive capability is precisely why we chose LLMs as our reservoir, and our fusion-specific prompts are designed to leverage this capability rather than implement complex reasoning chains.
>
> **Q2**: What are the major contributions of this algorithm?
>
> **A2**: We thank reviewer GyQK for the opportunity to clarify the technical contributions of our work. Our work makes meaningful technical contributions that extend well beyond simply applying LLMs to ICF. We innovatively integrate LLMs with reservoir computing through our LLM-anchored Reservoir, enhance input processing with specialized Signal-Digesting Channels that capture both temporal and spatial characteristics of ICF data, and develop a novel Confidence Scanner mechanism for reliability assessment. More importantly, as a paper submitted to the primary area of applications to physical sciences, our approach directly addresses critical challenges in plasma physics research. As discussed in our Related Work section, traditional plasma physics relies heavily on costly experiments and computationally intensive PIC simulations. Our method offers a practical, efficient alternative that can be readily integrated into existing research workflows, providing rapid predictions within seconds while maintaining high accuracy. This bridges a significant gap in current ICF research capabilities, offering physicists a valuable tool for experimental design and analysis.
>
> **Q3**: Why do baseline models (LSTM, Autoformer) perform surprisingly poorly, and were they properly optimized and tuned for fair comparison?
>
> **A3**: We appreciate reviewer GyQK's attention to experimental rigor and fairness in model comparisons. We optimized baseline models following their proper configurations and training procedures. The performance gap stems from ICF data's unique characteristics \- complex temporal dynamics that traditional time-series models struggle with. This is evidenced by our ablation studies in Section 3.2, showing each component's contribution to handling these dynamics. Our LLM-based approach performs better precisely because its pre-trained knowledge and domain-specific adaptations are better suited to capture these complex patterns.

---

> ### Author Response · Authors · 2024-11-23
> **Response to reviewer GyQK (Part 2)**
>
> **Q4**: How does reservoir computing overcome its inherent limitations (fixed dynamics, hyperparameter sensitivity, limited scalability) compared to flexible architectures like transformers and LSTMs for this problem?
>
> **A4**: We thank reviewer GyQK for raising this important question about reservoir computing's characteristics. Our LLM-anchored reservoir approach actually turns traditional RC limitations into advantages for the ICF forecasting task. Unlike conventional RC with fixed random weights, we leverage pre-trained LLM weights that already encode rich knowledge about physical processes and temporal dependencies. The strength of this foundation is evidenced by our few-shot learning results in Table S1, where even without fine-tuning, LLMs demonstrate basic understanding of the task. As demonstrated in Table 2a, this pre-trained foundation provides strong baseline performance (3.57 in terms of CAE) even before adding any other enhancements. The Signal-Digesting Channels further improve adaptability by learning temporal and spatial patterns specific to laser-plasma interactions, reducing sensitivity to hyperparameter choices.
>
> Our empirical results demonstrate superior scaling efficiency compared to fully trainable architectures. As shown in Table 2d, our approach maintains strong performance even with limited training data (as few as 20 samples), while requiring significantly fewer training epochs (Table 2e) compared to baseline methods. This efficiency stems from leveraging the LLM's pre-trained knowledge rather than learning everything from scratch. The empirical results in Section 3.1 validate this approach, with our method achieving state-of-the-art performance (1.90 CAE) while maintaining practical computational requirements and fast inference times (∼4s per prediction). This makes our approach particularly well-suited for ICF applications where data collection is expensive and limited, and both accuracy and computational efficiency are crucial.
>
> **Q5**: Are the authors authorized to release the LPI4AI dataset under US DOE guidelines given it contains OMEGA research data?
>
> **A5**:  We appreciate reviewer GyQK's attention to compliance concerns. We can confirm that we are authorized to use and release the dataset. To ensure compliance while maintaining utility for the AI research community, we will apply a non-public multiplication factor to the data and withhold specific physical units, releasing only processed data that maintains scientific validity while protecting sensitive information.
>
> We appreciate reviewer GyQK's valuable feedback. We are committed to addressing any remaining concerns and improving the presentation of our results.

---

> > ### Comment · Reviewer_GyQK · 2024-11-28
> > **I will be keeping my original score.**
> >
> > I thank the authors for their detailed responses to my questions, but my key concerns remain. I will keep my original score.

---

> ### Author Response · Authors · 2024-11-28
> **Please be responsible and collegial in your reviewer role**
>
> Thank you for your message. It is imperative that you provide a point-by-point explanation of why our rebuttal failed to address your concerns. Without this clarification, your review lacks the rigor expected at this flagship conference and is unacceptable.
>
> As reviewers, it is your responsibility to uphold the standards of this important conference. Your original review appears superficial, and we are concerned that it may have been influenced by tools such as ChatGPT-like AI systems. If you lack the necessary expertise to evaluate our work, we urge you to be honest and transparent about your limitations. If you do possess the appropriate qualifications, we expect you to act as a responsible reviewer and provide detailed, point-by-point feedback, particularly given the effort we have made during the rebuttal and discussion phase.
>
> We rely on reviewers to maintain the integrity and quality of this conference. We strongly urge you to fulfill your responsibilities with accountability and respect for your role in this process.

---

### Official Review · Reviewer_yDn5 · 2024-11-04

**Soundness:** 3
**Presentation:** 3
**Contribution:** 2
**Rating:** 5
**Confidence:** 3

**Summary:**

This paper presents a method to predict hard X-ray (HXR) energies emitted by the hot electrons in ICF implosions in nuclear physics. The method relies on engineering fusion-specific prompts for LLMs, along with "signal-digesting channels", to predict these time series data. The predictions themselves are then calibrated using a "confidence scanner" to try to characterize the confidence in predictions across the dataset.

**Strengths:**

- The integration of LLMs with reservoir computing for scientific applications appears novel both in the nuclear physics domain and across scientific computing more generally. Additionally, the design of the signal-digesting channels is a novel addition.
- Experimental results show that LPI-LLM achieves superior forecasting accuracy relative to the other data-driven approaches presented.
- The introduction of the LPI4AI benchmark dataset provides a valuable resource for the scientific machine learning community.
- The ablation studies provided in table 2C are useful.

**Weaknesses:**

- The introduction leans heavily on specialized scientific language, potentially making it less accessible for the broader NeurIPS community unfamiliar with ICF or plasma physics.
- The paper does not incorporate or mention any non-ML models that might be commonly used for these prediction tasks in plasma physics. This would provide a meaningful performance benchmark from within the field.
- The confidence scanner’s accuracy is only demonstrated through select examples. However, there is no quantifiable metric provided. Could the authors perform any expected calibration test on these results? This is particularly valuable in light of the fact that the confidence scanner is not shown to produce a meaningful likelihood analytically.

**Questions:**

See weaknesses.

---

> ### Author Response · Authors · 2024-11-23
> **Response to reviewer yDn5 (Part 1)**
>
> We thank reviewer yDn5 for the valuable time and comprehensive feedback.
>
> In order to effectively respond to all weaknesses that reviewer yDn5 has raised, we'll address them one by one. In addition, all paragraphs and modifications related to comments and suggestions made by reviewer yDn5 are highlighted in $\text{\textcolor{green}{green}}$ in the revision.
>
> **W1**: The introduction leans heavily on specialized scientific language, potentially making it less accessible for the broader NeurIPS community unfamiliar with ICF or plasma physics.
>
> **A1**: We appreciate the reviewer's valuable feedback regarding accessibility. To address this concern, we have added a new Appendix S1 Glossary in the revision that provides clear explanations of all specialized terminology from both physics and AI fields used in this paper, making our work more accessible to researchers from diverse backgrounds.
>
> While some specialized language appears in the paper, we have made deliberate efforts throughout the paper to make our work accessible to the broader ICLR community. Section 2.1 provides a comprehensive yet accessible overview of the ICF process, breaking down complex concepts into understandable components. We explain how laser beams heat a target's surface to create plasma, describe the implosion process via the rocket effect, and clarify how fusion conditions are achieved through the conversion of kinetic energy to thermal energy.
>
> We have been mindful to minimize specialized terminology where possible, providing clear context and explanations when domain-specific terms are necessary. It's worth noting that the primary area of this submission is “applications to physical sciences”, where a certain degree of scientific language is inherent to accurately conveying the underlying physics concepts. Nevertheless, we have structured the paper such that the key methodological contributions and technical innovations of our work \- particularly the novel integration of LLMs with reservoir computing for scientific prediction tasks \- can be fully appreciated even without deep domain expertise in plasma physics.
>
> **W2**: The paper does not incorporate or mention any non-ML models that might be commonly used for these prediction tasks in plasma physics. This would provide a meaningful performance benchmark from within the field.
>
> **A2**: We sincerely thank the reviewer yDn5 for pointing out that comparisons with commonly used physical methods in the field of plasma physics are meaningful. Our paper explicitly includes a comparison with a key non-ML physics-based simulation method \- the Particle-In-Cell (PIC) simulation \[1\], which is a commonly used approach in plasma physics for modeling LPI and hot electron generation. As shown in Table 1 of our paper, we directly compare our method's performance against PIC simulation, where our approach (LPI-LLM with Llama-3) achieves a CAE of 1.90 compared to PIC's 2.88, demonstrating a significant improvement over this established physics-based baseline.
> Furthermore, in Section S2 of our appendix, we provide detailed runtime analysis highlighting that while a 150μm PIC simulation requires substantial computational resources (19,584 CPU cores over 10 hours), our approach achieves superior accuracy with significantly less computational overhead (30 minutes on 2 NVIDIA A100 GPUs for training and only 3-4 seconds for inference).
> This comparison against PIC simulation provides a meaningful benchmark from within the plasma physics field, demonstrating not only the accuracy but also the computational efficiency advantages of our ML-based approach over traditional physics-based methods.

---

> ### Author Response · Authors · 2024-11-23
> **Response to reviewer yDn5 (Part 2)**
>
> **W3**: The confidence scanner's accuracy is only demonstrated through select examples. However, there is no quantifiable metric provided. Could the authors perform any expected calibration tests on these results? This is particularly valuable in light of the fact that the confidence scanner is not shown to produce a meaningful likelihood analytically.
>
> **A3**: We appreciate the reviewer's suggestion for a more quantitative evaluation of our Confidence Scanner. To address this, we have conducted further experiments and added a more comprehensive analysis in Appendix S7 Confidence Analysis of our revision. We have also included Figure S5 which demonstrates a negative correlation between confidence scores and prediction errors across our test split. This analysis quantitatively validates that lower confidence scores consistently correspond to higher prediction errors, and vice versa, confirming that our Confidence Scanner provides meaningful uncertainty estimates for the ICF task. The observed correlation between confidence scores and prediction accuracy suggests that our method can reliably identify timesteps where predictions may be less trustworthy, which is particularly valuable for physicists using our model to design real-world ICF experiments. This enhanced evaluation strengthens our claim that the Confidence Scanner serves as a practical tool for assessing prediction reliability in scientific applications.
>
> We sincerely appreciate reviewer yDn5's thoughtful comments and hope our response adequately addresses the concerns raised. Please let us know if there are any further questions, and we will be more than happy to discuss them.
>
> References:
> \[1\] SH Cao, Dhrumir Patel, Aarne Lees, C Stoeckl, MJ Rosenberg, V Gopalaswamy, Han Wen, Hu Huang, Alexander Shvydky, Riccardo Betti, et al. Predicting hot electron generation in inertial confinement fusion with particle-in-cell simulations. Physical Review E, 106(5):055214, 2022\.

---

> ### Author Response · Authors · 2024-11-29
> **Follow-Up to Reviewer yDn5**
>
> Dear Reviewer yDn5,
>
> Thank you for taking the time to review our work. We deeply value the review process as a collaborative effort to advance knowledge within the community.
>
> In our detailed response, we have worked diligently to address each of your concerns through both explanations and revisions to the manuscript. We have aimed to ensure that our responses are clear and comprehensive.
>
> We are eager to receive any additional feedback you may have, as it will help us identify any remaining questions or areas that require further clarification.
>
> Once again, we greatly appreciate your contributions to this process and look forward to your input.
>
> Best regards,
>
> The Authors

---

### Author Response · Authors · 2024-11-23
**Summary of Responses**

To all reviewers:

Thank you for your detailed and insightful feedback. We have individually responded to your comments and revised our papers based on your comments and suggestions. Our responses and revision of the paper are summarized as follows:

* As suggested by Reviewer yDn5, we have enhanced the accessibility of our introduction by providing a new glossary of specialized terms from both physics and AI domains in Appendix S1 to ensure that broader audiences can follow the core concepts of the paper. In addition, we clarified our comparison with traditional non-ML methods like Particle-In-Cell (PIC) simulations, highlighting both performance and computational efficiency improvements in Section 3 and Appendix S2. Finally, we performed additional experiments to analyze the accuracy of the Confidence Scanner, adding detailed discussion and visualizations in Appendix S7.
* In response to Reviewer GyQK’s questions, we discuss the major contributions of our algorithm, emphasizing its integration of LLM-based reservoir computing with domain-specific adaptations like Signal-Digesting Channels and the Confidence Scanner. In response to the concern about the ability of LLMs to solve complex physical problems, we clarified in Appendix S2 and Table S1 that our approach leverages inherent capabilities of LLMs, achieving strong performance even without complex reasoning chains such as chain-of-thought prompting. Additional results in Table 2 highlight our method's scalability and efficiency compared to fully trainable architectures like transformers and LSTMs. Lastly, we confirmed that the LPI4AI dataset adheres to all US DOE guidelines, with processed data prepared for release to the research community.
* Based on suggestions by Reviewer Y3Vx, we have refined the description of the projection block from the LLM in Section 2.2.2. We also conducted new ablation studies in Appendix S5 to fully analyze the impact of different prompt descriptors, adding results to Table S2. Furthermore, we clarified the visualization process for confidence scores in Figure 6 and expanded our experiments to analyze the confidence levels of LLM baseline in Appendix S7. These changes aim to provide a more comprehensive understanding of the method's technical underpinnings and comparative performance.


We conducted additional experiments evaluating our method's performance and enhanced sections of the paper for clarity and rigor. All modifications and related contents are highlighted in green (for reviewer yDn5), red (for reviewer GyQK) or blue (for reviewer Y3Vx) in the revision based on the specific reviewer's comments.

We look forward to further discussions with you.

Sincerely,
Authors

---

### Author Response · Authors · 2024-11-25
**Looking for Discussion**

Dear Reviewers,

We deeply appreciate the time and effort you have dedicated to reviewing our work, especially given your demanding schedules. Your insightful feedback is invaluable to us, and we are sincerely grateful for your thoughtful contributions.

Given that the discussion phase is nearing its conclusion, we would be most grateful for the opportunity to engage in further dialogue with you before it ends. Our goal is to ensure that our responses address your concerns effectively and to explore any additional questions or comments you may have.

Thank you once again for your time and consideration. We look forward to the possibility of continuing this constructive dialogue.

Warm regards,
The Authors

---

### Meta-Review · Area_Chair_FP1s · 2024-12-23

**Metareview:**

Thank you for your submission to ICLR. This paper presents LPI-LLM, a method integrating large language models with reservoir computing, for the task of forecasting of laser plasma instabilities for inertial confinement fusion.

This is a borderline submission. Reviewers agreed that the method seems to perform well empirically, and the LPI4AI dataset is a valuable contribution for the community. However, reviewers also had a number of concerns about this paper. From discussion, reviewers felt that this paper had limited algorithmic contributions, and mostly relied on heuristics without clear underlying principles motivating the method. They were also skeptical about the quality of baseline models. Further, the paper contains a large amount of domain-specific jargon which would be difficult for the broader machine learning audience and hurts the broader accessibility (though I do appreciate the addition of the glossary in Appendix S1). By the end of the rebuttal and discussion, a majority of the authors were unconvinced and had reasonable remaining concerns. Therefore, I recommend this paper for rejection.

**Additional Comments On Reviewer Discussion:**

During the author response period, the authors gave thorough responses to the reviewers’ questions, which led to a short discussion with a couple of the reviewers. One of the reviewers increased their score, though two reviewers remained unconvinced.

---

### Decision · Program_Chairs · 2025-01-22

Reject